# Optimal Transfer Learning for Missing Not-at-Random Matrix Completion

**Akhil Jalan** [1]  **Yassir Jedra** [2]  **Arya Mazumdar** [3]  **Soumendu Sundar Mukherjee** [4]  **Purnamrita Sarkar** [5]

## Abstract

We study transfer learning for matrix completion in a Missing Not-at-Random (MNAR) setting that is motivated by biological problems. The target matrix $Q$ has entire rows and columns missing, making estimation impossible without side information. To address this, we use a noisy and incomplete source matrix $P$, which relates to $Q$ via a feature shift in latent space. We consider both the *active* and *passive* sampling of rows and columns. We establish minimax lower bounds for entrywise estimation error in each setting. Our computationally efficient estimation framework achieves this lower bound for the active setting, which leverages the source data to query the most informative rows and columns of $Q$. This avoids the need for *incoherence* assumptions required for rate optimality in the passive sampling setting. We demonstrate the effectiveness of our approach through comparisons with existing algorithms on real-world biological datasets.

## 1. Introduction

We study transfer learning in the context of matrix completion, a fundamental problem motivated by theory (Candès and Recht, 2009; Candès and Tao, 2010) and practice (Fernández-Val et al., 2021; Einav and Cleary, 2022; Gao et al., 2022).

A major body of work studies matrix completion in the Missing Completely-at-Random (MCAR) setting (Jain et al., 2013; Chatterjee, 2015; Chen et al., 2020), where each entry is observed i.i.d. with probability $p$. A more general missingness pattern, known as Missing Not-at-Random (MNAR), considers an underlying *propensity matrix* $p_{ij}$ so that the $(i,j)^{th}$ entry is observed independently with probability $p_{ij}$ (Ma and Chen, 2019; Bhattacharya and Chatterjee, 2022). Various MNAR models have been formulated based on missingness structures in panel data (Agarwal et al., 2023b), recommender systems (Jedra et al., 2023), and electronic health records (Zhou et al., 2023).

Motivated by biological problems, we consider a challenging MNAR structure where most rows and columns of $\widetilde{Q}$ (a noisy version of $Q$) are entirely missing. Specifically, we consider both the *active sampling* and *passive sampling* settings for $\widetilde{Q}$. In active sampling, a practitioner can choose rows $R$ and columns $C$ so that entries in $R \times C$ are observed. This follows experimental design constraints in metabolite balancing experiments (Christensen and Nielsen, 2000a), marker selection for single-cell RNA sequencing (Vargo and Gilbert, 2020), patient selection for companion diagnostics (Huber et al., 2022), and gene expression microarrays (Hu et al., 2021).

The requirement that entire rows and columns must be observed is due to real-world constraints, such as the use of certain assays or experimental protocols. For example, *metabolite balancing* is a method for measuring pairwise metabolic interactions in cells, but requires choosing a set of metabolites (rows & columns) beforehand (Christensen and Nielsen, 2000b). Another example comes from gene expression microarray measurements, which require a choice of patients (rows) and genes (columns) to measure beforehand (Hu et al., 2021). We study both of these settings in Section 3.

In the *passive sampling* setting, the practitioner cannot choose the experiments. We model this by sampling each row (column) with probability $p_{\text{Row}}$ ($p_{\text{Col}}$). For example, microarray analysis detects RNA segments corresponding to known genes by using chemical hybridization. However, rows may be missing because of a patient sample failing to hybridize, and columns may be missing because of gene probe failure (Hu et al., 2021). For an illustration, see Figure 1.

This setting is inherently difficult because there are many entries $(i,j)$ for which row $i$ and column $j$ are *both* missing in $\widetilde{Q}$. Clearly, even when $Q$ is low-rank and incoherent, estimation is impossible without side information (Proposition 2.1). Transfer learning is *necessary* to achieve vanishing estimation error since no information about $Q_{ij}$ is known. Hence, we consider transfer learning in a setting where one has a noisy and masked $\widetilde{P}$ corresponding to a source matrix $P$. $P$

---

[1]Department of Computer Science, UT Austin, USA [2]Laboratory for Information & Decision Systems (LIDS), MIT, USA [3]Halıcıoğlu Data Science Institute & Department of Computer Science and Engineering, UC San Diego, USA [4]Statistics and Mathematics Unit (SMU), Indian Statistical Institute, Kolkata, India [5]Department of Statistics and Data Sciences, UT Austin, USA. Correspondence to: Akhil Jalan <akhiljalan@utexas.edu>.

*Proceedings of the 42$^{nd}$ International Conference on Machine Learning*, Vancouver, Canada. PMLR 267, 2025. Copyright 2025 by the author(s).

and $Q$ are related by a distribution shift in their latent singular subspaces (Definition 1.2), which is a common model in e.g. Genome-Wide Association Studies (McGrath et al., 2024) and Electronic Health Records (Zhou et al., 2023).

**Contributions.** Below, we list our contributions:

(i) We obtain **minimax lower bounds** for entrywise estimation error for both the active (Theorem 2.2) and passive sampling settings (Theorem 2.12).

(ii) We give a **computationally efficient** estimation framework for both sampling settings. Our procedure is **minimax optimal** for the active setting (Theorem 2.6). We also establish minimax optimality for the passive setting under *incoherence* assumptions (Theorem 2.9).

(iii) We compare the performance of our algorithm with existing algorithms on **real-world datasets** for gene expression microarrays and metabolic modeling (Section 3).

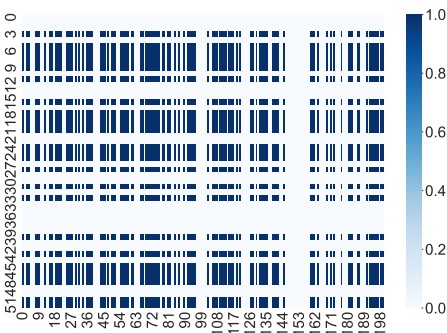

*Figure 1.* The missingness matrix for gene expression levels on Day 2 of a sepsis study (Parnell et al., 2013) shows entire rows (patients) and columns (genes) as missing, due to e.g. probe-target hybridization failure of the Illumina HT-12 gene expression microarray (Hu et al., 2021). We mark missing entries as 0 (white) and present entries as 1 (blue). This motivates our missingness model (Eq. (1) and Eq. (2)).

**Setup.** $P, Q \in \mathbb{R}^{m \times n}$ are the underlying source and target matrices, related by a distributional shift in their latent singular subspaces (Definition 1.2). We observe a noisy and possibly masked $\widetilde{P}$. The observation model of $\widetilde{Q}$ depends on which setting below we consider. We will introduce both observation models here, and discuss the estimation framework used for both models in Section 2.2.

*(i) Active Sampling Setting.* We have a budget of $T_{\text{row}}$ rows and $T_{\text{col}}$ columns. We select rows $i_1, ..., i_{T_{\text{row}}}$ and columns $j_1, ..., j_{T_{\text{col}}}$, possibly at random, and with repeats allowed. Let $n_{ij} \geq 0$ be the number of times *both*

row $i$ and column $j$ are chosen. Then, we have $n_{ij}$ independent noisy observations $\widetilde{Q}_{i,j}^{(1)}, ..., \widetilde{Q}_{i,j}^{(n_{ij})}$ such that:

$$\tilde{Q}_{i,j}^{(t)} = \begin{cases} Q_{ij} + \zeta_{i,j}^{(t)} & \text{if } n_{ij} > 0, \\ \star & \text{otherwise,} \end{cases} \quad (1)$$

For $\zeta_{i,j}^{(t)} \overset{i.i.d}{\sim} \mathcal{N}(0, \sigma_Q^2)$.

*(ii) Passive Sampling Setting.* Instead of row and column budgets, there are probabilities $p_{\text{Row}}, p_{\text{Col}} \in [0, 1]$ corresponding to the random row mask $\eta_1, ..., \eta_m \overset{i.i.d.}{\sim} \text{Ber}(p_{\text{row}})$ and column mask $\nu_1, ..., \nu_n \overset{i.i.d}{\sim} \text{Ber}(p_{\text{col}})$. Entry $(i, j)$ of $Q$ is noisily observed if $\eta_i = \nu_j = 1$, and missing otherwise.

$$\tilde{Q}_{ij} = \begin{cases} Q_{ij} + \zeta_{i,j} & \text{if } \eta_i = \nu_j = 1, \\ \star & \text{otherwise,} \end{cases} \quad (2)$$

where $\zeta_{i,j} \overset{i.i.d}{\sim} \mathcal{N}(0, \sigma_Q^2)$.

## 1.1. Organization of the Paper

We give our main theoretical findings, including lower and upper bounds for the active and passive sampling settings, in Section 2. Next, we compare our methods against existing algorithms on real-world and synthetic datasets in Section 3. Finally, we discuss related work in Section 4 and conclusions in Section 5.

## 1.2. Notation and Problem Setup

We use lowercase letters $a, b, c$ to denote (real) scalars, boldface $\boldsymbol{x}, \boldsymbol{y}, \boldsymbol{z}$ to denote vectors, and uppercase $A, B, C$ to denote matrices. For $n \geq 1$, let $[n] := \{1, ..., n\}$, $I_n$ be the identity matrix and $(\boldsymbol{e}_i)_{i=1}^n$ the canonical basis vectors. Let $a \vee b := \max\{a, b\}$ and $a \wedge b := \min\{a, b\}$. For multisets $S, T$ and $A \in \mathbb{R}^{m \times n}$, let $A[S, T] \in \mathbb{R}^{|S| \times |T|}$ be the submatrix with row and column indices in $S, T$ respectively, possibly with repeated entries from $A$. Let $\otimes$ denote the tensor (Kronecker) product: for $A \in \mathbb{R}^{m \times n}, B \in \mathbb{R}^{s \times t}$, $(A \otimes B) \in \mathbb{R}^{ms \times nt}$ with $(A \otimes B)_{i(r-1)+v, j(s-1)+w} = A_{ij} B_{vw}$. We denote the Frobenius norm as $\|A\|_F$, max norm as $\|A\|_{\max} := \max_{i,j} |A_{ij}|$, and $2 \to \infty$ norm as $\|A\|_{2 \to \infty} := \max_i \|A^T \boldsymbol{e}_i\|_2$. Asymptotics $O(\cdot), o(\cdot), \Omega(\cdot), \omega(\cdot)$ are with respect to $m \wedge n$ unless specified otherwise. Recall that, for integer $n, d$ such that $d \leq n$, the *Stiefel manifold* $\mathcal{O}^{n \times d}$ (Hatcher, 2002) consists of all $U \in \mathbb{R}^{n \times d}$ such that $U^T U = I_d$.

We now define matrix incoherence, which measures how concentrated the entries of the singular vectors are.

**Definition 1.1** (Incoherence)**.** Let $M$ be an $m \times n$ matrix of rank $d$, and write its SVD as $M = U \Sigma V^\top$. The left (resp. right) incoherence parameter of $M$ is defined as $\mu_U = m\|U\|_{2 \to \infty}^2 / d$ (resp. $\mu_V = n\|V\|_{2 \to \infty}^2 / d$). The incoherence parameter of $M$ is defined as $\mu(M) := \max\{\mu_U, \mu_V\}$.

We now formally define the distribution shift from $P$ to $Q$, which generalizes the latent space rotation model (Xu et al., 2013; McGrath et al., 2024).

**Definition 1.2** (Matrix Transfer Model). In the matrix transfer model, we have source and target matrices $P, Q \in \mathbb{R}^{m \times n}$ such that:

*(i)* (Low-Rank) Let $P = U_P \Sigma_P V_P^\top$ for some $d \leq m \wedge n$ where $U_P \in \mathcal{O}^{m \times d}, V_P \in \mathcal{O}^{n \times d}$, and $\Sigma_P \succeq 0$ is diagonal $d \times d$.

*(ii)* (Distribution shift) There exist $T_1, T_2, R \in \mathbb{R}^{d \times d}$ such that $Q = U_P T_1 R T_2^T V_P^T$, and $\|T_i\|_2 = O(1)$ for $i = 1, 2$.

We will define the parameter space as:

$$\mathcal{F}_{m,n,d} = \Big\{ (P,Q) \in \mathbb{R}^{m \times n} \times \mathbb{R}^{m \times n} : P = U \Sigma_P V^T,$$

$$Q = U T_1 R T_2^T V^T, U \in \mathcal{O}^{m \times d}, V \in \mathcal{O}^{n \times d},$$

$$T_1, T_2, R \in \mathbb{R}^{d \times d}, \Sigma_P \succeq 0 \Big\} \quad (3)$$

Definition 1.2 requires that the $d$-dimensional features of rows and columns lie in a shared subspace for $P, Q$. Consider the matrix of associations between $m$ genetic variants (e.g. the MC1R gene) and $n$ phenotypes (e.g. dark hair) for different populations $P, Q$ (e.g. England and Spain) (McGrath et al., 2024). The above model ensures that the latent feature vector for a genotype (resp. phenotype) in $Q$ is a linear combination of those in $P$.

Note that $T_1, T_2$ are not necessarily rotations and can even be singular. We set $\|T_i\|_2 = O(1)$ to simplify theorem statements, but it is not required.

## 2. Main Findings

We first show that without transfer – side information from the source data $P$ – completing the target matrix $Q$ is impossible. To this end, we present a minimax lower bound on the expected prediction error. First, we define the parameter space of matrices with bounded incoherence:

$$\mathcal{T}_{mn}^{(d)} = \Big\{ Q \in \mathbb{R}^{m \times n} : \text{rank}(Q) \leq d,$$

$$\mu(Q) \leq O\big(\log(m \vee n)\big) \Big\}. \quad (4)$$

**Proposition 2.1** (Minimax Error of MNAR Matrix Completion Without Transfer). *Let $m, n \geq 1$ and $d \leq m \wedge n$. Let $\Psi = (Q, \sigma, p_{\text{Row}}, p_{\text{Col}})$ where $Q \in \mathcal{T}_{mn}^{(d)}$, $\sigma^2 > 0$, and $p_{\text{Row}}, p_{\text{Col}} \in [0,1]$. Let $\mathbb{P}_\Psi$ denote the law of the random matrix $\widetilde{Q}$ defined as in Eq. (2) with $\sigma_Q = \sigma$, and denote the expectation under this law as $\mathbb{E}_\Psi$. The minimax rate of estimation is:*

$$\inf_{\hat{Q}} \sup_{\substack{Q \in \mathcal{T}_{mn}^{(d)} p_{\text{Row}} \leq .99 \\ p_{\text{Col}} \leq .99}} \inf_\Psi \mathbb{E}\left[ \frac{1}{mn} \|Q - \hat{Q}\|_F^2 \right] \geq \Omega(d\sigma^2).$$

An immediate consequence of the above proposition is that the minimax rate for max squared error $\|\hat{Q} - Q\|_{\max}^2$ is also $\Omega(d\sigma^2)$. We see that in both error metrics, vanishing estimation error is impossible without transfer learning.

### 2.1. Lower Bound for Active Sampling Setting

We now give a minimax lower bound for $Q$ estimation in the active sampling setting.

**Theorem 2.2** (Minimax Lower Bound for $Q$-estimation with Active Sampling). *Fix $m, n$ and $2 \leq d \leq m \wedge n$. Fix $\sigma^2 > 0$ and let $|\Omega| = T_{\text{row}} \cdot T_{\text{col}}$.*

*Let $\mathbb{P}_{P,Q,\sigma^2}$ be the distribution of $(\widetilde{P}, \widetilde{Q})$ where $\widetilde{P} := P$ and $\widetilde{Q} := Q + G$ where $G_{ij} \overset{i.i.d}{\sim} N(0, \sigma^2)$.*

*Let $\mathcal{Q}$ be the **class of estimators** which observe $\widetilde{P}$, and choose row and column samples according to the budgets $T_{\text{row}}, T_{\text{col}}$ as in Eq. (1), and then return some estimator $\hat{Q} \in \mathbb{R}^{m \times n}$. Then, there exists absolute constant $C > 0$ such that minimax rate of estimation is:*

$$\inf_{\hat{Q} \in \mathcal{Q}} \sup_{(P,Q) \in \mathcal{F}_{m,n,d}} \mathbb{E}_{\mathbb{P}_{P,Q,\sigma^2}}[\|\hat{Q} - Q\|_{\max}^2] \geq \frac{Cd^2\sigma^2}{|\Omega|}.$$

We prove Theorem 2.2 using a generalization of Fano's method (Verdú et al., 1994). We construct a family of distributions indexed by $d^2$ source/target pairs $(P^{(s)}, Q^{(s)})_{s=1}^{d^2}$. The source $P$ is the same for all $s$, while each pair of target matrices $Q^{(s)}, Q^{(s')}$ differs in at most 2 entries. For example, say entries $(5,6)$ and $(8,7)$ are different between $Q^{(1)}$ and $Q^{(2)}$. Regardless of the choice of row/column samples, the *average* KL divergence of a pair of targets is small. If e.g. the entries $(5,6), (8,7)$ are heavily sampled, then the estimator can distinguish $Q^{(1)}, Q^{(2)}$ well, but cannot distinguish $Q^{(t)}, Q^{(t')}$ for all $t, t'$ pairs that are equal on $(5,6)$ and $(8,7)$.

### 2.2. Estimation Framework

Next, we describe our estimation framework. Given $\widetilde{P}$ and $\widetilde{Q}[R,C]$, where $R, C$ can come from either the active (Eq. (1)) or passive sampling (Eq. (2)) setting, we estimate $\hat{Q}$ via the least-squares estimator.

**Least Squares Estimator.**

1. Extract features via SVD from $\tilde{P} = \widehat{U}_P \widehat{\Sigma}_P \widehat{V}_P^T$.

2. Let $\Omega$ be the multiset of observed entries. Then solve

$$\hat{\Theta}_Q := \arg\min_{\Theta \in \mathbb{R}^{d \times d}} \sum_{(i,j) \in \Omega} |\widetilde{Q}_{ij} - \hat{\boldsymbol{u}}_i^\top \Theta \hat{\boldsymbol{v}}_j|^2, \quad (5)$$

where $\hat{\boldsymbol{u}}_i := \hat{U}_P^T \boldsymbol{e}_i, \hat{\boldsymbol{v}}_j := \hat{V}_P^T \boldsymbol{e}_j$.

3. Estimate $\hat{Q}$:

$$\hat{Q}_{ij} = \hat{\boldsymbol{u}}_i^\top \hat{\Theta}_Q \hat{\boldsymbol{v}}_j. \tag{6}$$

This fully specifies $\hat{Q}$ in the passive sampling setting (Eq. (2)). For the active sampling setting, we must also specify how rows and columns are chosen.

Active sampling poses two main challenges. First, it is not clear how to leverage $\widetilde{P}$ for sampling $\widetilde{Q}$ because samples are chosen *before* observing $\widetilde{Q}$, so the distribution shift from $P$ to $Q$ is unknown. Second, the best design depends on the choice of estimator and vice versa.

Surprisingly, we show that for the right choice of experimental design, the optimal estimator is precisely the least-squares estimator $\hat{Q}$ as in Eq. (6). We use the classical $G$-optimal design (Pukelsheim, 2006), which has been used in reinforcement learning to achieve minimax optimal exploration (Lattimore and Szepesvári, 2020b) and optimal policies for linear Markov Decision Processes (Taupin et al., 2023).

**Definition 2.3** ($\epsilon$-approximate $G$-optimal design)**.** Let $\mathcal{A} \subset \mathbb{R}^d$ be a finite set. For a distribution $\pi : \mathcal{A} \to [0,1]$, its $G$-value is defined as

$$g(\pi) := \max_{\boldsymbol{a} \in \mathcal{A}} \left[ \boldsymbol{a}^T \left( \sum_{\boldsymbol{a} \in \mathcal{A}} \pi(\boldsymbol{a}) \boldsymbol{a} \boldsymbol{a}^T \right)^{-1} \boldsymbol{a} \right].$$

For $\epsilon > 0$, we say $\hat{\pi}$ is $\epsilon$-approximately $G$-optimal if

$$g(\hat{\pi}) \leq (1+\epsilon) \inf_\pi g(\pi).$$

If $\epsilon = 0$, we say $\hat{\pi}$ is simply $G$-optimal.

Notice that in Eq. (5), the covariates are tensor products $(\hat{\boldsymbol{v}}_j \otimes \hat{\boldsymbol{u}}_i)$ of column and row features. The $G$-optimal design is useful because it respects the tensor structure of the least-squares estimator. We prove this via the Kiefer-Wolfowitz Theorem (Lattimore and Szepesvári, 2020b).

**Proposition 2.4** (Tensorization of $G$-optimal design)**.** *Let $U \in \mathbb{R}^{m \times d_1}, V \in \mathbb{R}^{n \times d_2}$. Let $\rho$ be a $G$-optimal design for $\{U^T \boldsymbol{e}_i : i \in [m]\}$ and $\zeta$ be a $G$-optimal design for $\{V^T \boldsymbol{e}_j : j \in [n]\}$. Let $\pi(i, j) = \rho(i)\zeta(j)$ be a distribution on $[m] \times [n]$. Then $\pi$ is a $G$-optimal design on $\{V^T \boldsymbol{e}_j \otimes U^T \boldsymbol{e}_i : i \in [m], j \in [n]\}$.*

Consider a maximally coherent $P$ that is nonzero at entry $(3,5)$ and zero elsewhere. Then $Q$ is also zero outside $(3,5)$. By the Kiefer-Wolfowitz Theorem, the $G$-optimal design for rows (resp. columns) samples row 3 (resp. column 5) with probability 1. So, if $\widetilde{P}$ is not too noisy, then the $G$-optimal design samples *precisely the useful rows/columns*.

In light of Proposition 2.4, we leverage the tensorization property to sample rows and columns as follows.

**Active Sampling.** Given $\hat{U}, \hat{V}$, and budget $T_\text{row}, T_\text{col}$,

1. Compute $\epsilon$-approximate $G$-optimal designs $\hat{\rho}, \hat{\zeta}$ for $\{\hat{U}_P^T \boldsymbol{e}_i : i \in [m]\}$ and $\{\hat{V}_P^T \boldsymbol{e}_j : j \in [n]\}$ respectively, with the Frank-Wolfe algorithm (Lattimore and Szepesvári, 2020b).

2. Sample $i_1, ... i_{T_\text{row}} \overset{i.i.d}{\sim} \hat{\rho}$ and $j_1, ... j_{T_\text{col}} \overset{i.i.d}{\sim} \hat{\zeta}$.

Finally, we specify the assumption we need on the source data $\widetilde{P}$, called Singular Subspace Recovery (SSR).

**Assumption 2.5** ($\epsilon$-SSR)**.** Given $\widetilde{P} \in (\mathbb{R} \cup \{\star\})^{m \times n}$, we have access to a method that outputs estimates $\hat{U}_P \in \mathcal{O}^{m \times d}$ and $\hat{V}_P \in \mathcal{O}^{n \times d}$, such that:

$$\inf_{W_U \in \mathcal{O}^{d \times d}} \|\hat{U} - U W_U\|_{2 \to \infty} \leq \epsilon_\text{SSR},$$
$$\text{and} \quad \inf_{W_V \in \mathcal{O}^{d \times d}} \|\hat{V} - V W_V\|_{2 \to \infty} \leq \epsilon_\text{SSR} \tag{7}$$

for some $\epsilon_\text{SSR} > 0$.

This assumption holds for a number of models. For instance, recent works in both MCAR (Chen et al., 2020) and MNAR (Agarwal et al., 2023b; Jedra et al., 2023) settings give estimation methods for $\hat{P}$ with entry-wise error bounds. In Appendix A.2, we prove that these entry-wise guarantees, combined with standard theoretical assumptions such as incoherence, imply Assumption 2.5.

We now give our main upper bound, stated in terms of max squared error. Note that our upper bound for max error immediately implies upper bounds for commonly used metrics including mean squared (Frobenius) error, root mean squared error, and mean absolute error.

**Theorem 2.6** (Generic error bound for active sampling)**.** *Let $\hat{Q}$ be the active sampling estimator with $T_\text{row}, T_\text{col} \geq 20d\log(m+n)$. Then, for absolute constants $C, C' > 0$, and all $\epsilon < \frac{1}{10}$,*

$$\mathbb{P}_{\widetilde{P}, \widetilde{Q}} \left[ \|\hat{Q} - Q\|_\text{max}^2 \leq C(1+\epsilon) \left( \frac{d^2 \sigma_Q^2 \log(m+n)}{|T_\text{col}||T_\text{row}|} \right. \right.$$
$$\left. \left. + d^2 \epsilon_\text{SSR}^2 \|Q\|_2^2 \right) \right]$$
$$\geq 1 - C'(m+n)^{-2}.$$

We will discuss implications of Theorem 2.6 in Remark 2.7. First, we give some intuition. Notice that Theorem 2.6 (and Theorem 2.9) gives an error bound as a sum of two terms, which depend on the sample size and $\epsilon_\text{SSR}$ respectively. To see why, let $\Omega$ be the set of observed entries, either in a passive or active sampling setting. Let $\hat{\boldsymbol{u}}_i, \hat{\boldsymbol{v}}_j$ be the covariates

as in Eq. (5). The observation $\widetilde{Q}_{ij}$ can be decomposed:

$$
\begin{aligned}
\widetilde{Q}_{ij} &= Q_{ij} + (\widetilde{Q}_{ij} - Q_{ij}) \\
&= \widehat{\boldsymbol{u}}_i^\top \Theta_Q \widehat{\boldsymbol{v}}_j + \underbrace{\epsilon_{ij}}_{\text{misspecification } \widetilde{P}} + \underbrace{(\widetilde{Q}_{ij} - Q_{ij})}_{\text{noise}} \quad (8)
\end{aligned}
$$

The population estimand $\Theta_Q \in \mathbb{R}^{d \times d}$, which is estimated in Eq. (5), is:

$$
\Theta_Q := W_U^T T_1 R T_2^T W_V,
$$

where $T_1$, $T_2$ are the distribution shift matrices as in Definition 1.2, and $W_U, W_V \in \mathcal{O}^{d \times d}$ are some rotations. The misspecification error is due to the estimation error of the singular subspaces of $P$ and depends on $\epsilon_{\text{SSR}}$ as follows:

$$
\begin{aligned}
\epsilon_{ij} := &\, \boldsymbol{e}_i^T (\hat{U} - U W_U) \Theta_Q \hat{V} \boldsymbol{e}_j \\
&+ \boldsymbol{e}_i^T \hat{U} \Theta_Q (\hat{V} - V W_V) \boldsymbol{e}_j \\
&+ \boldsymbol{e}_i^T (\hat{U} - U W_U) \Theta_Q (\hat{V} - V W_V) \boldsymbol{e}_j
\end{aligned}
$$

Therefore $\epsilon_{ij}^2 = O(\epsilon_{\text{SSR}}^2 \|Q\|_2^2)$ for all $i, j$.[1] Notice the misspecification error is independent of the estimator $\widehat{\Theta}_Q$, so it will not depend on sample size. This explains the appearance of the two summands in our upper bounds. The first term depends on estimation error $\Theta_Q - \hat{\Theta}_Q$, which is unique to the sampling method. The second depends on misspecification, which is common to both.

**Remark 2.7** (Minimax Optimality for MNAR and MCAR Source Data). The rate of Theorem 2.6 is minimax-optimal in the usual transfer learning regime when target data is noisy ($\sigma_Q$ large) and limited ($|\Omega| := |T_{\text{row}}||T_{\text{col}}|$ small).

Suppose $P$ is rank $d$, $\mu$-incoherent, with singular values $\sigma_1 \geq \cdots \geq \sigma_d$, condition number $\kappa$ and $m = n$. For the MNAR $\widetilde{P}$ setting, suppose each $\widetilde{P}_{ij}$ has i.i.d. additive noise $\mathcal{N}(0, \sigma_P^2)$ with sampling sparsity factor $n^{-\beta}$ for $\beta \in [0,1]$ and $\sigma_P = O(1)$. By (Jedra et al., 2023), $\hat{Q}$ is minimax-optimal if

$$
\frac{4\mu^3 d^3 \kappa^2 \|Q\|_2^2}{n^{1 + \frac{2-\beta}{d}}} \lesssim \frac{\sigma_Q^2}{|\Omega|},
$$

where $\lesssim$ ignores $\log(m+n)^{O(1)}$ factors. For the MCAR $\widetilde{P}$ setting, suppose $\widetilde{P}$ has additive noise $\mathcal{N}(0, \sigma_P^2)$ and observed entries i.i.d. with probability $p \gtrsim \frac{\kappa^4 \mu^2 d^2}{n}$, with $\sigma_P \sqrt{\frac{n}{p}} \lesssim \frac{\sigma_d(P)}{\sqrt{\kappa^4 \mu d}}$. Letting $|\Omega| = n^2 p_{\text{Row}} p_{\text{Col}}$, by (Chen et al., 2020), $\hat{Q}$ is minimax-optimal if

$$
\frac{\mu^6 d^4 \|Q\|_2^2}{n^2} \lesssim \frac{\sigma_Q^2}{|\Omega|}.
$$

While the results of (Jedra et al., 2023; Chen et al., 2020) used in Remark 2.7 require incoherence, recent work also gives guarantees on $\epsilon_{\text{SSR}}$ without incoherence assumptions, although in limited settings.

**Remark 2.8** (Incoherence-free minimax optimality). Let $P \in \mathbb{R}^{n \times n}$ be rank-1 and Hermitian, and $\widetilde{P} = P + W$ where $W$ is Hermitian with i.i.d. $\mathcal{N}(0, \sigma_P^2)$ noise on the upper triangle. Under the assumptions of (Yan and Levin, 2024), for constant $C > 0$, $\hat{Q}$ is minimax optimal if

$$
\frac{C \sigma_P^2 (\log n)^{O(1)} \|Q\|_2^2}{\|P\|_2^2} \leq \frac{\sigma_Q^2}{|\Omega|}.
$$

Taking $|\Omega| = O(\log n)$ since $d = 1$, and $\|Q\|_2 = O(\|P\|_2)$, we require

$$
C \sigma_P^2 (\log n)^{O(1)} \leq \sigma_Q^2.
$$

### 2.3. Passive Sampling

We next give the estimation error for the passive sampling setting. The rate almost exactly matches Theorem 2.6, but we pay an extra factor due to incoherence. This is because unlike the active sampling setting, if $\ell_2$ mass of the features is highly concentrated in a few rows and columns, then the passive sample will simply miss these with constant probability. To give a high probability guarantee, we require that features cannot be too highly concentrated.

**Theorem 2.9** (Generic Error Bound for $\hat{Q}$). *Let $\hat{Q}$ be as in Eq. (6) and $C > 0$ an absolute constant. Suppose $P$ has left/right incoherence $\mu_U, \mu_V$ respectively, and $p_{\text{Row}}, p_{\text{Col}}$ are such that $\frac{p_{\text{Row}} m}{C d \log m} \geq \mu_U + \frac{\epsilon_{\text{SSR}}^2 m}{d}$, $\frac{p_{\text{Col}} n}{C d \log n} \geq \mu_V + \frac{\epsilon_{\text{SSR}}^2 n}{d}$. Let $\mu = \mu_U \mu_V$. Then*

$$
\begin{aligned}
\mathbb{P}\Big[ \|\hat{Q} - Q\|_{\max}^2 &\leq C\mu \Big( \frac{d^2 \sigma_Q^2 \log(m+n)}{p_{\text{Row}} p_{\text{Col}} mn} \\
&\quad + d^2 \epsilon_{\text{SSR}}^2 \|Q\|_2^2 \Big) \Big] \\
&\geq 1 - O((m \wedge n)^{-2}).
\end{aligned}
$$

If $P$ is coherent, the sample complexity $|\Omega| \approx p_{\text{Row}} p_{\text{Col}} mn$ needed to achieve vanishing estimation error in Theorem 2.9 may be large. By contrast, our active sampling with $G$-optimal design requires only $|\Omega| \gtrsim d^2 \sigma_Q^2$ (Theorem 2.6). This shows the advantage of active sampling, which can query the most informative rows/columns when $P$ is coherent.

### 2.4. Lower Bound for Passive Sampling

We give a lower bound for the passive sampling setting in terms of a fixed, arbitrary mask. To exclude degenerate cases such as all entries being observed, we require the following definition.

---

[1] In fact $\epsilon_{ij}^2 = O(\epsilon_{\text{SSR}}^2 \|R\|_2^2)$, but we report bounds with the weaker $O(\epsilon_{\text{SSR}}^2 \|Q\|_2^2)$ for ease of reading.

**Definition 2.10** (Nondegeneracy). Let $p > 0$ and $\eta_1, ..., \eta_m \overset{i.i.d}{\sim} \text{Ber}(p)$. Let $D \in \{0,1\}^{m \times m}$ be diagonal with $D_{ii} = \eta_i$. We say $(\eta_i)_{i=1}^m$ is $p$-nondegenerate for $U \in \mathcal{O}^{n \times d}$ if $\left| \|DU\|_2 - \sqrt{p} \right| \leq \frac{\sqrt{p}}{10}$.

The Matrix Bernstein inequality (Chen et al., 2021) implies that masks are nondegenerate with high probability.

**Proposition 2.11.** *Under the conditions of Theorem 2.9, the event that both $(\eta_i)_{i=1}^m$ is $p_{\text{Row}}$-nondegenerate for $\hat{U}_P$ and that $(\nu_j)_{j=1}^n$ is $p_{\text{Col}}$-nondegenerate for $\hat{V}_P$ holds with probability $\geq 1 - 2(m \wedge n)^{-10}$.*

We can now state our lower bound, proved via Fano's method.

**Theorem 2.12** (Minimax Lower Bound for Passive Sampling). *Let $\mathcal{F}_{m,n,d}$ be the parameter space of Theorem 2.2. Let*

$$\mathcal{G}_{m,n,d} := \left\{ (P,Q) \in \mathcal{F}_{m,n,d} : P,Q \text{ are } O(1) - \text{incoherent} \right\}$$

*Suppose $(\eta_i)_{i=1}^m, (\nu_j)_{j=1}^n$ are nondegenerate with respect to $U, V$ respectively. Let $\mathbb{P}_{Q,\sigma^2,p_{\text{Row}},p_{\text{Col}}}$ be the law of the random matrix $\widetilde{Q}$ generated as in Eq. (2) with $\sigma = \sigma_Q$.*

*There exists absolute constant $C > 0$ such that minimax rate of estimation is:*

$$\inf_{\hat{Q}} \sup_{(P,Q) \in \mathcal{G}_{m,n,d}} \mathbb{E}_{\mathbb{P}_{Q,\sigma^2,p_{\text{Row}},p_{\text{Col}}}} \left[ \frac{1}{mn} \|\hat{Q} - Q\|_F^2 \,\middle|\, (\eta_i)_{i=1}^m, \right.$$
$$\left. (\nu_j)_{j=1}^n \right] \geq \frac{Cd^2 \sigma_Q^2}{p_{\text{Row}} p_{\text{Col}} mn}$$

*We immediately obtain the same lower bound for max squared error.*

We see that our error rate for passive sampling in Theorem 2.9 is minimax-optimal when $\mu = O(1)$, modulo bounds on $\epsilon_{\text{SSR}}$ as in Remark 2.7.

Unlike the lower bound for max squared error in active sampling (Theorem 2.2), Theorem 2.12 gives a lower bound for the mean-squared error, which is strictly stronger. An interesting question is whether Theorem 2.12 can be generalized to incoherence greater than a constant. We leave this for future work.

# 3. Experiments

In this section, we compare both our active and passive sampling estimators against existing methods on real-world and simulated datasets.

**Experimental setup.** We compare against two baselines from the matrix completion literature. First, we use the MNAR matrix completion method of (Bhattacharya and

*Table 1.* Summary of real-world datasets. The $2 \to \infty$ norms are for $U_P, V_P, U_Q, V_Q$ respectively. Notice these are within $[0,1]$ always, and $2 \to \infty$ norm of 1 implies maximal coherence.

| DATASET | SHAPE | RANK | $2 \to \infty$ NORMS |
|---|---|---|---|
| GENE EXPR. | $31 \times 300$ | 4 | 0.55, 0.30, 0.64, 0.38 |
| METABOLIC | $251 \times 251$ | 8 | 0.99, 0.99, 0.99, 0.99 |

Chatterjee, 2022). We tune the method by passing in the true rank of $Q$ as well as the rank of the mask matrix. Second, we use the transfer learning method of (Levin et al., 2022b). This method is designed for matrix completion, but in a missingness structure different from our MNAR setting. For shorthand, we will refer to these as *BC22* and *LLL22* respectively. See Appendix B for precise details of our implementations. Additionally, see Appendix B.1 for comparison to a VAE baseline from (Ipsen et al., 2021).

The input to each of these, as well as our passive sampling method, is the pair $\widetilde{P}, \widetilde{Q}$. The method of (Bhattacharya and Chatterjee, 2022) requires input matrices to have entries in $[-1, 1]$ so we normalize all $\widetilde{P}, \widetilde{Q}$ by their maximum entry in absolute value, for all methods. We also compute the active sampling estimator by fixing the budgets $T_{\text{row}} = m \cdot p_{\text{Row}}, T_{\text{col}} = n \cdot p_{\text{Col}}$ throughout.

## 3.1. Real World Experiments

In this section we study real-world datasets on gene expression microarrays in a whole-blood sepsis study (Parnell et al., 2013), and weighted metabolic networks of gram-negative bacteria (King et al., 2016). Table 1 summarizes the datasets, and Appendix B gives more details on our data preparation.

**Patient Gene Expression Matrices.** The matrices $P, Q$ represent the gene expression for patients in a sepsis study (Parnell et al., 2013). Here $P, Q \in \mathbb{R}^{31 \times 300}$ where $P_{ij}$ measures the expression level of gene $j$ in patient $i$ on day 1 of the study, and $Q$ corresponds to day 2 of the study.

Figure 2 displays the maximum squared error for a range of masking probabilities on $\widetilde{Q}$. We see that both active and passive sampling perform well even at small sample sizes, while the transfer baseline method (Levin et al., 2022b) achieves a worse but nontrivial maximum error.

Notably, active sampling is no better than passive sampling here. This makes sense because $P, Q$ are relatively incoherent (Table 1), so our theoretical guarantees are the same.

In fact, active sampling displays higher variation in error, due to the variability in random sampling from the $G$-optimal design. It is known that the $G$-optimal design for any $\mathcal{A} \subset \mathbb{R}^d$ has support size $O(d^2)$ (Lattimore and Szepesvári, 2020a), so the sampled set of rows and columns will vary somewhat

from one experiment to the next.

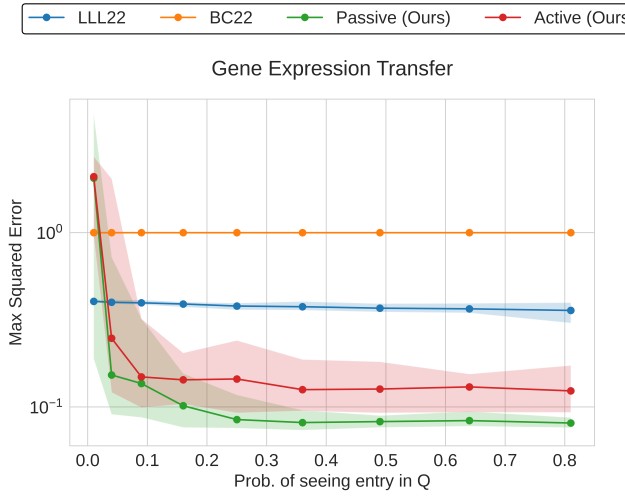

Figure 2. Max-squared error of $\hat{Q} - Q$. Here, $\widetilde{Q}$ has $p_{\text{Row}} = p_{\text{Col}}$ varying along the $x$-axis, which displays $p_{\text{Row}}^2$. We set $\sigma_Q = 0.1$, and $P$ is fully observed. For each method, we show the median of the errors across 50 independent runs, as well as the [10,90] percentile.

**Weighted Metabolic Network Adjacency Matrices.** We collect weighted metabolic networks from the BiGG Genome Scale Metabolic Models repository (King et al., 2016), consistent with recent work on transfer learning for network estimation (Jalan et al., 2024). Specifically, $P, Q \in \mathbb{R}^{251 \times 251}$ where $P_{ij} \geq 0$ counts the number of co-occurrences of metabolites $i$ and $j$ in a reaction for organism $P$. $Q_{ij}$ represents the same quantity in a different organism $Q$. We use the gram-negative bacteria *E. coli W* and *P. putida* for $P,Q$ respectively. Unlike (Jalan et al., 2024), we do not need to truncate the adjacency matrices to $\{0,1\}$, allowing us to handle edge weights. This makes a difference, because without truncation the edge weights distribution is highly skewed for both $P,Q$ (see Appendix B).

Figure 3 shows max squared error for a range of masking probabilities on $\widetilde{Q}$. We see that active sampling does well, while passive sampling is very poor (note however, that passive sampling does relatively well for mean-squared error - Figure 12). This is because $P,Q$ are almost maximally coherent (Table 1), so the assumptions of our guarantee for passive sampling (Theorem 2.9) do not hold. By contrast, active sampling performs well even in this highly coherent setting.

### 3.2. Simulations

In this section, we further probe the effects of incoherence by testing on two highly coherent synthetic datasets (described below). Table 2 displays our results, with $p_{\text{Row}} = p_{\text{Col}} = 0.1, \sigma_Q = 0.1$, and $P$ fully observed. Note that $0.1 \approx \frac{2d\log n}{n}$ here, so $p_{\text{Row}}, p_{\text{Col}}$ are near the theoretical limit

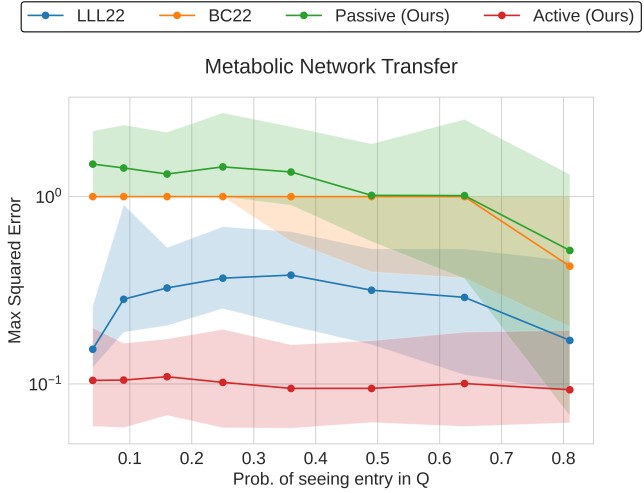

Figure 3. Max-squared error of $\hat{Q} - Q$, with the same experimental parameters as Figure 2.

of our guarantees even for incoherent matrices.

Each table entry shows $\hat{\mu} \pm 2\hat{\sigma}$ for mean-squared error across 50 independent trials. We find that for a stylized example of maximally coherent $P,Q$, active sampling is much better than all other methods. However, for less stylized $P,Q$ that are still not incoherent, active and passive sampling are comparable, and outperform both baselines.

**Stylized Coherent Model.** For $n = 200, d = 5$ we generate $U_P, V_P \in \{0,1\}^{n \times d}$ via $(U_P)_{ii} = 1$, $(V_P)_{(n-i),i} = 1.0$ and the other entries zero. We sample the diagonal entries of $\Sigma_P, \Sigma_Q \in \mathbb{R}^{d \times d}$ iid uniformly at random from $[0.5,1]$. Then $P = U_P \Sigma_P V_P^T$ and $Q = U_P \Sigma_Q V_P^T$. We call this class "Coherent."

**Matrix Partition Model.** For a less stylized class, let $m = 300, n = 200, d = 5, a = 0.1, b = 0.8$. We generate partitions $U_P \in \{0,1\}^{m \times d}, V_P \in \{0,1\}^{n \times d}$ where each row is uniformly at random from $\{e_1,...,e_d\}$. Then, $B_P \in [0,1]^{d \times d}$ is generated by sampling $C \in [0,1]^{d \times d}$ with $C_{ij} \overset{i.i.d}{\sim} \text{Unif}([0,b])$ and $(B_P)_{ij} = C_{ij} + \mathbf{1}_{i=j} a$. Finally, we sample permutations $\Pi_1, \Pi_2 \in \{0,1\}^{d \times d}$ uniformly at random from all such permutations. Then, $P = U_P B_P V_P^T$ and $Q = U_P \Pi_1 B_P \Pi_2^T V_P^T$. We call this class "Matrix Partition Model" in analogy with the Planted Partition Model (Abbe, 2017). Spectral arguments show that such matrices are somewhat coherent (Lee et al., 2014), although not maximally so.

### 3.3. Ablation Studies

Our main focus is to understand how sample budgets $T_{\text{row}}, T_{\text{col}}$, or probabilities $p_{\text{Row}}, p_{\text{Col}}$ affects the estimation error for transfer learning. We also perform ablation studies

*Table 2.* Comparison of the errors of different approaches on synthetic data.

|  | COHERENT | PARTITION |
|---|---|---|
| PASSIVE (OURS) | $0.084 \pm 0.039 \times 10^{-3}$ | $\mathbf{0.040 \pm 0.090}$ |
| ACTIVE (OURS) | $\mathbf{0.009 \pm 0.015} \times 10^{-3}$ | $0.046 \pm 0.074$ |
| LLL22 | $0.061 \pm 0.037 \times 10^{-3}$ | $0.134 \pm 0.011$ |
| BC22 | $0.789 \pm 0.644 \times 10^{-3}$ | $0.305 \pm 0.002$ |

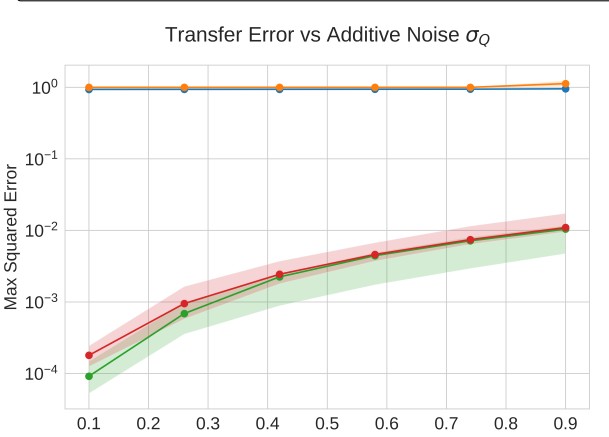

*Figure 4.* Ablation study for the effect of additive target noise in the Matrix Partition Model. For each method, we display the median max-squared error across 10 independent runs, as well as the [10,90] percentile.

to test the effect of other model parameters, such as rank, dimension, noise variance, etc. Figure 4 shows the effect of target noise variance on maximum error in the Matrix Partition Model with $m = 300, n = 200, d = 5, a = 0.1, b = 0.8, p_{\text{Row}} = 0.5, p_{\text{Col}} = 0.5$. Due to space constraints, we defer our additional ablation studies to Appendix B.

## 4. Related Work

We review the most relevant literature here. For additional discussion, we refer to the surveys (De Handschutter et al., 2021; Jafarov, 2022) for matrix completion and (Zhuang et al., 2019; Kim et al., 2022) for transfer learning.

**Matrix Completion.** Most matrix completion algorithms require a Missing Completely at Random (MCAR) assumption (Candès and Recht, 2009; Chatterjee, 2015; Davenport et al., 2014; Zhong et al., 2019), where each $Q_{ij}$ is observed with probability $p$ independently of all others. The Missing Not-at-Random setting allows the masking probability of $Q_{ij}$ to depend on the value of $Q_{ij}$ itself (Ma and Chen, 2019; Bhattacharya and Chatterjee, 2022; Jedra et al., 2023),

but still assumes that entries are masked independently of one another. If masking variables are dependent, then authors assume identifiability of the matrix conditioned on the masking (Agarwal et al., 2023b), or that entries in every row and column are observed (Simchowitz et al., 2023). By contrast, we study one of the simplest possible MNAR models in which entries of $\widetilde{Q}$ are *not* independent and entire rows and columns can be missing. This MNAR model is motivated by biological problems (Christensen and Nielsen, 2000a; Hu et al., 2021; Einav and Cleary, 2022).

**Transfer learning.** Transfer learning has been well-studied in learning theory (Ben-David et al., 2006; Cortes et al., 2008; Crammer et al., 2008). Recent works address various supervised learning (Reeve et al., 2021; Cai and Wei, 2021b; Ma et al., 2023; Cai and Pu, 2024) and unsupervised learning settings (Gu et al., 2024; Ding and Ma, 2024). Statistical works consider minimax rates of estimation, and computationally efficient estimators to achieve such rates (Tripuraneni et al., 2020; Agarwal et al., 2023a; Cai and Wei, 2021a; Ma et al., 2023; Cody and Beling, 2023; Cai and Pu, 2024). In applications, transfer learning from data-rich to data-poor domains has applications in biostatistics (Kshirsagar, 2015; Datta et al., 2021), epidemiology (Apostolopoulos and Bessiana, 2020), computer vision (Tzeng et al., 2017; Neyshabur et al., 2020), language models (Han et al., 2021), and other areas.

Transfer learning for matrix completion typically assumes the source $P$ and target $Q$ are observed in an MCAR fashion, and are related through a rotation in latent space (Xu et al., 2013; McGrath et al., 2024; He et al., 2024). Rotational shift is a special case of our distribution shift model (Definition 1.2), which allows for any linear shift in latent space. On the other hand, works that study transfer learning for specific classes of matrices typically assume distributional shifts that are unique to those structures, such as in latent variable networks (Jalan et al., 2024) or the log-linear word production model (Zhou et al., 2023).

**Optimal experimental design.** Choosing a set of maximally informative experiments is a classical problem in statistics (Smith, 1918; Pukelsheim, 2006) with connections to active learning (Dasgupta, 2011), bandits (Abbasi-Yadkori et al., 2011), and reinforcement learning (Lattimore et al., 2020). Optimal designs have been studied for domain adaptation (Rai et al., 2010; Xie et al., 2022), misspecified regression (Lattimore et al., 2020), and linear Markov Decision Processes (Jedra et al., 2023). In our active sampling setting, we *jointly* query rows and columns to observe the corresponding submatrix of $\widetilde{Q}$, rather than one entry at a time (Chakraborty et al., 2013; Ruchansky et al., 2015; Bhargava et al., 2017). But, the optimal row queries depend on column queries (and vice versa) – so we use the tensorization property of $G$-optimal designs (Proposition 2.4) to prove global optimality with respect to joint row/column samplers.

# 5. Conclusion and Future Work

We study transfer learning for a challenging MNAR model of matrix completion. We obtain minimax lower bounds for entrywise estimation of $Q$ in both the active (Theorem 2.2) and passive sampling settings (Theorem 2.12). We give a computationally efficient minimax-optimal estimator that uses tensorization of $G$-optimal designs in the active setting (Theorem 2.6). Further, in the passive setting, we give a rate-optimal estimator under incoherence assumptions (Theorem 2.9). Finally, we experimentally validate our findings on data from gene expression micoarrays and metabolic modeling.

Future work could consider even more difficult missingness structures, such as when the masks $(\eta_i)_{i=1}^m$, $(\nu_j)_{j=1}^n$ are dependent. If the mask can be partitioned into subsets whose mutual dependencies are small, an Efron-Stein argument (Paulin et al., 2016) may work. Is bounded dependence necessary? Moreover, one can consider other kinds of side information, such as gene-level features in Genome-Wide Association Studies (McGrath et al., 2024). Finally, there can be other interesting nonlinear models for transfer between source and target matrices.

## Impact Statement

This paper presents work whose goal is to advance the field of Machine Learning. There are many potential societal consequences of our work, none which we feel must be specifically highlighted here.

## Acknowledgments

AJ and PS gratefully acknowledge NSF grants 2217069, 2019844, and DMS 2109155. YJ is supported by the Knut and Alice Wallenberg Foundation Postdoctoral Scholarship Program under grant KAW 2022.0366. AM was supported by NSF awards 2217058 and 2133484. SSM was partially supported by an INSPIRE research grant (DST/INSPIRE/04/2018/002193) from the Dept. of Science and Technology, Govt. of India, a Start-Up Grant from Indian Statistical Institute, and a Prime Minister Early Career Research Grant (ANRF/ECRG/2024/006704/PMS) from the Anusandhan National Research Foundation, Govt. of India.

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

# A. Proofs and Additional Results

## A.1. Preliminaries

We will repeatedly make use of the vectorization operator.

**Definition A.1** (Vectorization). For $X \in \mathbb{R}^{n \times d}$, the vectorization $\text{vec}(X) \in \mathbb{R}^{nd}$ is the vector whose first $n$ entries correspond to the first column of $X$, and next $n$ entries correspond to the second column of $X$, and so on.

We can vectorize matrix products as follows.

**Lemma A.2** ((Horn and Johnson, 2012)). *Let $A, B, X$ be matrices of shapes such that $AXB$ is well-defined. Then:*

$$vec(AXB) = (B^T \otimes A)vec(X).$$

## A.2. From Entrywise Guarantees to SSR

We prove that Assumption 2.5 follows from entrywise estimation guarantees on the source.

**Proposition A.3.** *Let $P$ an $m \times n$ matrix of rank $r$. Let $\epsilon > 0$, and $\widehat{P}$ be a rank-$r$ estimate of $P$, satisfying*

$$\|\widehat{P} - P\|_{\max} \leq \epsilon \|P\|_{\max}. \tag{9}$$

*Consider the SVDs $P = U\Sigma V^\top$, and $\widehat{P} = \widehat{U}\widehat{\Sigma}\widehat{V}^\top$. Then, it holds that*

$$\min_{W \in \mathcal{O}^{r \times r}} \|U - \widehat{U}R\|_{2 \to \infty}$$
$$\leq \frac{(2\sqrt{n} + (2+\sqrt{2})\sqrt{mn}\|UU^\top\|_{2\to\infty})\|P - \widehat{P}\|_{\max}}{\sigma_r(P)}$$
$$\min_{W \in \mathcal{O}^{r \times r}} \|V - \widehat{V}W\|_{2 \to \infty} \leq$$
$$\leq \frac{(2\sqrt{m} + (2+\sqrt{2})\sqrt{mn}\|VV^\top\|_{2\to\infty})\|P - \widehat{P}\|_{\max}}{\sigma_r(P)}$$

*provided that $\sqrt{mn}\epsilon\|P\|_{\max} \leq \frac{\sigma_r(P)}{2}$.*

Below, we give a result showing that entry-wise guarantees imply subspace recovery in the two-to-infinity guarantee.

*Proof.* We will only prove the result concerning the left subspaces $U$ and $\widehat{U}$. Our first step is to relate the errors $\widehat{U}R - U$ and $UU^\top\widehat{U} - U$. We will introduce in our computations the sign matrix[2] of $U^\top\widehat{U}$, namely $\text{sgn}(U^\top\widehat{U})$ which is a rotation matrix. We have

$$\min_{W \in \mathcal{O}^{r \times r}} \|UW - \widehat{U}\|_{2\to\infty} \leq \|U\text{sgn}(U^\top\widehat{U}) - \widehat{U}\|_{2\to\infty}$$
$$\leq \|U(U^\top\widehat{U}) - \widehat{U}U\|_{2\to\infty} + \|U\|_{2\to\infty}\|U^\top\widehat{U} - \text{sgn}(U^\top\widehat{U})\|_{\text{op}}.$$

Moreover, we also know (e.g., see Lemma 4.15 (Chen et al., 2021)) that

$$\|\widehat{U}^\top U - \text{sgn}(\widehat{U}^\top U)\|_{\text{op}} \leq \|\sin(\Theta)\|_{\text{op}},$$

and using the Theorem Davis-Kahan we obtain

$$\|\widehat{U}^\top U - \text{sgn}(\widehat{U}^\top U)\|_{\text{op}} \leq \|\sin(\Theta)\|_{\text{op}} \leq \frac{\sqrt{2}\|M - \widehat{M}\|_{\text{op}}}{\sigma_r(M)}.$$

Thus, we conclude that

$$\min_{W \in \mathcal{O}^{r \times r}} \|UW - \widehat{U}\|_{2\to\infty} \leq \|U(U^\top\widehat{U}) - \widehat{U}U\|_{2\to\infty} + \frac{\sqrt{2}\|U\|_{2\to\infty}\|M - \widehat{M}\|_{\text{op}}}{\sigma_r(M)}. \tag{10}$$

---

[2]The sign matrix of an $n \times n$ matrix $Z$ with SVD $U_Z\Sigma_Z V_Z^\top$ is given by $\text{sgn}(Z) = U_Z V_Z^\top \in \mathcal{O}^{n \times n}$.

Next, we show that $\min_{W \in \mathcal{O}^{r \times r}} \|UW - \widehat{U}\|_{2\to\infty}$ can be well controlled by the error $M - \widehat{M}$. On the one hand, we have triangular inequality, and noting that $UU^\top M = M$ and $\widehat{U}\widehat{U}^\top \widehat{M} = \widehat{M}$ that

$$\|(UU^\top - \widehat{U}\widehat{U}^\top)\widehat{M}\|_{2\to\infty} \leq \|UU^\top M - \widehat{U}\widehat{U}^\top \widehat{M}\|_{2\to\infty} + \|UU^\top(M - \widehat{M})\|_{2\to\infty}$$
$$\leq \|M - \widehat{M}\|_{2\to\infty} + \|UU^\top\|_{2\to\infty}\|M - \widehat{M}\|_{\text{op}}$$

On the other hand, we have

$$\|(UU^\top - \widehat{U}\widehat{U}^\top)\widehat{M}\|_{2\to\infty} = \|(U(U^\top\widehat{U}) - \widehat{U})\widehat{\Sigma}\widehat{V}^\top\|_{2\to\infty}$$
$$= \|(U(U^\top\widehat{U}) - \widehat{U})\widehat{\Sigma}\|_{2\to\infty}$$
$$\geq \|U(U^\top\widehat{U}) - \widehat{U}\|_{2\to\infty}\sigma_r(\widehat{M})$$
$$\geq \|U(U^\top\widehat{U}) - \widehat{U}\|_{2\to\infty}\sigma_r(M) - \|U(U^\top\widehat{U}) - \widehat{U}\|_{2\to\infty}\|M - \widehat{M}\|_{\text{op}},$$

where in the last inequality we used Weyl's inequality: $|\sigma_r(M) - \sigma_r(\widehat{M})| \leq \|M - \widehat{M}\|_{\text{op}}$. We combine the above inequalities to obtain

$$\|U(U^\top\widehat{U}) - \widehat{U}\|_{2\to\infty} \leq \frac{\|M - \widehat{M}\|_{2\to\infty} + \|UU^\top\|_{2\to\infty}\|M - \widehat{M}\|_{\text{op}} + \|U(U^\top\widehat{U}) - \widehat{U}\|_{2\to\infty}\|\widehat{M} - M\|_{\text{op}}}{\sigma_r(M)}$$

If the following condition holds

$$\|M - \widehat{M}\|_{\text{op}} \leq \sqrt{mn}\|M - \widehat{M}\|_{\max} \leq \frac{\sigma_r(M)}{2},$$

then

$$\|U(U^\top\widehat{U}) - \widehat{U}\|_{2\to\infty} \leq \frac{\|M - \widehat{M}\|_{2\to\infty} + \|UU^\top\|_{2\to\infty}\|M - \widehat{M}\|_{\text{op}}}{\sigma_r(M)} + \frac{1}{2}\|U(U^\top\widehat{U}) - \widehat{U}\|_{2\to\infty}$$

which in turn gives

$$\|U(U^\top\widehat{U}) - \widehat{U}\|_{2\to\infty} \leq \frac{2\|M - \widehat{M}\|_{2\to\infty} + 2\|UU^\top\|_{2\to\infty}\|M - \widehat{M}\|_{\text{op}}}{\sigma_r(M)} \tag{11}$$

In summary we conclude that

$$\min_{W \in \mathcal{O}^{r \times r}} \|UW - \widehat{U}\|_{2\to\infty} \leq \frac{2\|M - \widehat{M}\|_{2\to\infty} + (2+\sqrt{2})\|UU^\top\|_{2\to\infty}\|M - \widehat{M}\|_{\text{op}}}{\sigma_r(M)} \tag{12}$$

Using the inequalities

$$\|M - \widehat{M}\|_{2\to\infty} \leq \sqrt{n}\|M - \widehat{M}\|_{\max} \qquad \text{and} \qquad \|M - \widehat{M}\|_{\text{op}} \leq \sqrt{mn}\|M - \widehat{M}\|_{\max},$$

we can express our bounds as

$$\min_{W \in \mathcal{O}^{r \times r}} \|UW - \widehat{U}\|_{2\to\infty} \leq \frac{(2\sqrt{n} + (2+\sqrt{2})\sqrt{mn}\|UU^\top\|_{2\to\infty})\|M - \widehat{M}\|_{\max}}{\sigma_r(M)}. \tag{13}$$

$\square$

A simple calculation also gives the following.

**Proposition A.4.** *Suppose $\hat{U} \in \mathcal{O}^{m \times r}$ satisfies Assumption 2.5 with bound $\epsilon_{\text{SSR}}$, and the population incoherence is $\mu_U := \frac{m\|U\|_{2\to\infty}^2}{d}$. Then $\hat{U}$ is $\gamma$-incoherent for $\gamma \leq 2\mu_U + \frac{2\epsilon_{\text{SSR}}^2 m}{d}$.*

## A.3. Proof of Proposition 2.1

We require the following special case of Hoeffding's inequality.

**Lemma A.5.** *Let* $X_1,...,X_n \overset{i.i.d}{\sim} Bernoulli(p)$. *Then:*

$$\mathbb{P}\left[\left|\frac{1}{n}\sum_i X_i - p\right| \geq \sqrt{\frac{\log n}{n}}\right] \leq 2n^{-2}$$

The following concentration is standard.

**Lemma A.6.** *Let* $\boldsymbol{x} \sim S^{n-1}$. *Then:*

$$\mathbb{P}[\|\boldsymbol{x}\|_\infty \geq C\sqrt{\frac{\log n}{n}}] \leq 1 - O(n^{-1/2})$$

*Proof.* By Hoeffding's inequality,

$$\mathbb{P}\left[\left|\sum_i X_i - np\right| \geq t\right] \leq 2\exp\left(-\frac{2t^2}{n}\right)$$

Let $t = \sqrt{n\log n}$. The conclusion follows. $\qquad\square$

Finally, we require the following version of the Hanson-Wright inequality.

**Theorem A.7** ((Rudelson and Vershynin, 2013) Theorem 2.1). *Let* $A \in \mathbb{R}^{m \times n}$ *be fixed and* $\boldsymbol{x} \in \mathbb{R}^n$ *a random vector with i.i.d. mean zero entries with variance* $1$ *and* $\|\boldsymbol{x}_i\|_{\psi_2} \leq K$ *for all* $i$. *Then there exists constant* $c > 0$ *such that for any* $t > 0$,

$$\mathbb{P}\left[|\|A\boldsymbol{x}\|_2 - \|A\|_F| > t\right] \leq 2\exp\left(-\frac{ct^2}{K^4\|A\|^2}\right)$$

We are ready to state our lower bound.

*Proof of Proposition 2.1.* Let $\boldsymbol{u}_1,...,\boldsymbol{u}_d \in \mathbb{R}^m$ be generated with iid $N(0,\frac{1}{m})$ entries and $\boldsymbol{v}_1,...,\boldsymbol{v}_d \in \mathbb{R}^m$ be generated with iid $N(0,\frac{1}{n})$ entries. Let $Q = \sum_{i=1}^d \boldsymbol{u}_i\boldsymbol{v}_i^T$.

We first analyze the incoherence of $Q$. We analyze the left-incoherence. Fix $i \in [m]$ and let $\boldsymbol{y} = (U^T\boldsymbol{e}_i)$. Then we apply Theorem A.7 with $\boldsymbol{x} = \sqrt{m}\boldsymbol{y}$ and $A = V$, to obtain that $\|A\boldsymbol{x}\| = \|\sqrt{m}VU^T\boldsymbol{e}_i\| \leq \|V\|_F + C'K^2\|V\|_2\sqrt{\log n}$ with probability $\geq 1 - n^{-10}$ for absolute constant $C' > 0$. Since $\boldsymbol{x}$ has iid $N(0,1)$ entries, the Orlicz norm constant is at most $K \leq 2$. Taking a union bound over all $i$, it follows that:

$$\mathbb{P}\left[\|\sqrt{m}VU^T\|_{2\to\infty} \leq \|V\|_F + 4C'\|V\|_2\sqrt{\log n}\right] \geq 1 - O(n^{-9})$$

It follows that the left incoherence is at most $O(\log n)$ with high probability. An identical application of Theorem A.7 with $A = U$ implies that the right-incoherence is at most $O(\log m)$. Let $\mathcal{E}'$ be the event that $Q$ is $O(\log(n \vee m))$ incoherent. Let $Q$ be the random matrix generated as above, conditioned on $\mathcal{E}'$. Note that $\mathbb{P}[\mathcal{E}'] \geq 1 - o(1)$.

Next, let $I \subset [m], J \subset [n]$ be the rows and columns of $Q$ that are seen in $\widetilde{Q}$. Then by Lemma A.5, $|I| \leq 0.99m + \sqrt{m\log m}$ and $|J| \leq 0.99n + \sqrt{n\log n}$ with probability $\geq 1 - 2n^{-2} - 2m^{-2}$. Let $\mathcal{E}$ be the event that the bounds on $I$ and $J$ both hold.

Consider $k \in [m] \setminus I, \ell \in [n] \setminus J$. None of the entries of $Q$ in the $k^{th}$ row or $\ell^{th}$ column are seen. Therefore, since $m - |I| \geq \Omega(m)$ and $n - |J| \geq \Omega(n)$, and since $\mathbb{P}[\mathcal{E}'] \geq 1 - o(1)$, there exists a constant $C$ such that for all $i \in [d]$, $Var(\boldsymbol{u}_{i;k}\boldsymbol{v}_{i;\ell}|\widetilde{Q}) \geq C$. Therefore, since $\boldsymbol{u}_1,...,\boldsymbol{u}_d,\boldsymbol{v}_1,...,\boldsymbol{v}_d$ are independent, for any $\hat{Q}$, we have:

$$\mathbb{E}[(\hat{Q}_{k\ell} - Q_{k\ell})^2|\widetilde{Q}] \geq Var(Q_{k\ell}|\widetilde{Q})$$

$$\geq \sum_{i=1}^d Var(\boldsymbol{u}_{i;k}\boldsymbol{v}_{i;\ell}|\widetilde{Q})$$

$$\geq Cd$$

Therefore, if we condition on $\mathcal{E}$, then $|[m]\setminus I|\geq\Omega(m)$ and $|[n]\setminus J|\geq\Omega(n)$, so $\mathbb{E}[\frac{1}{mn}\|\hat{Q}-Q\|_F^2|\widetilde{Q}]\geq cd$ for a constant $c>0$. Since $1-2n^{-2}-2m^{-2}\geq\frac{1}{2}$, we conclude that:

$$\mathbb{E}[\frac{1}{mn}\|\hat{Q}-Q\|_F^2|\widetilde{Q}]\geq\frac{1}{2}\mathbb{E}[\frac{1}{mn}\|\hat{Q}-Q\|_F^2|\widetilde{Q},\mathcal{E}]$$
$$\geq\frac{cd}{2}$$

$\square$

## A.4. Proof of Theorem 2.2

We require a version of Fano's theorem given in Theorem 7 of (Verdú et al., 1994).

**Theorem A.8** (Generalized Fano). *Let $\mathcal{P}$ be a family of probability measures, $(\mathcal{D},d)$ a metric space, and $\theta:\mathcal{P}\to\mathcal{D}$ a map that extracts the parameters of interest. Let $\mathcal{H}\subset\mathcal{P}$ be a finite subset of size $M$. Suppose $\alpha>0$ is such that for any distinct $H_i,H_j\in\mathcal{H}$,*

$$d(\theta(H_i),\theta(H_j))\geq\alpha.$$

*And, suppose that $\beta>0$ is such that:*

$$\log 2+\frac{1}{M^2}\sum_{i=1}^{M}\sum_{j=1}^{M}KL(H_i,H_j)\leq\beta\log M.$$

*Then,*

$$\inf_{\hat{\theta}}\sup_{P\in\mathcal{P}}\mathbb{E}[d(\theta(P),\hat{\theta})]\geq\alpha(1-\beta).$$

We also require a standard expression for the KL divergence of a pair of multivariate Gaussians.

**Lemma A.9.** *Let $\boldsymbol{\mu},\boldsymbol{\mu}'\in\mathbb{R}^d$ be distinct and $\Sigma\succ 0$. The KL divergence of two multivariate Gaussians sharing the same covariance is given as:*

$$KL(\mathcal{N}(\boldsymbol{\mu},\Sigma),\mathcal{N}(\boldsymbol{\mu}',\Sigma))=(\boldsymbol{\mu}-\boldsymbol{\mu}')^T\Sigma^{-1}(\boldsymbol{\mu}-\boldsymbol{\mu}')$$

We now prove our lower bound.

*Proof of Theorem 2.2.* Let $U\in\mathbb{R}^{m\times d},V\in\mathbb{R}^{n\times d}$ be such that $U_{ii}=1$ and $V_{ii}=1$ for $i\in[d]$, and all other entries are zero. Let $P=UV^T$. We construct a hypothesis space $\mathcal{H}=\{(P^{(ij)},Q^{(ij)}:i,j\in[d]\}$ of size $d^2$ where $P^{(ij)},Q^{(ij)}\in\mathbb{R}^{m\times n}$ as follows. For all members $ij$, we set $P^{(ij)}=P$. Next, let $R^{(ij)}=\gamma e_i e_j^T$ for $\gamma>0$ to be specified later. We set $Q^{(ij)}=UR^{(ij)}V^T$.

First, notice for any $(r,s)\neq(i,j)$ that:

$$\|Q^{(ij)}-Q^{(rs)}\|_{\max}^2=\gamma^2$$

Next, consider the KL divergences between a pair of hypotheses. Let $(\widetilde{P}^{(ij)},\widetilde{Q}^{(ij)})$ be the distribution of the data under hypothesis $(P^{(ij)},Q^{(ij)})$. Since $\widetilde{P}^{(ij)}=P^{(ij)}=P$ for all $(i,j)$, we must simply bound $KL(\widetilde{Q}^{(ij)},\widetilde{Q}^{(rs)})$ for each pair $(ij,rs)$. Now, let $\pi_R^{(ij)},\pi_C^{(ij)}$ be the row and column sampling distributions (possibly deterministic) respectively, based on the source data $\widetilde{P}^{(ij)}$. Since $\widetilde{P}^{(ij)}=P^{(ij)}=P$ for all $(i,j)$ we know that there is a pair of distributions $\pi_R,\pi_C$ such that $\pi_R^{(ij)}=\pi_R,\pi_C^{(ij)}=\pi_C$ for all $(i,j)$. In other words the sampling cannot depend on the hypothesis index $(i,j)$.

Next, we analyze $KL(\widetilde{Q}^{(ij)},\widetilde{Q}^{(rs)})$. Each distribution depends on the randomness of $\pi_R,\pi_C$ as well as the Gaussian noise. Let $R,C$ be the random multisets of rows and columns generated by $\pi_R,\pi_C$ according to the prescribed row/column budgets. By the chain rule for KL divergences (Theorem 2.15 of (Polyanskiy and Wu, 2024)), we have:

$$KL(\widetilde{Q}^{(ij)},\widetilde{Q}^{(rs)})=\mathbb{E}_{R,C}\left[KL\left((\widetilde{Q}^{(ij)}|R,C),(\widetilde{Q}^{(rs)}|R,C)\right)\right]$$

Note that the marginal term involving $\pi_R^{(ij)}, \pi_C^{(ij)}$ versus $\pi_R^{(rs)}, \pi_C^{(rs)}$ is zero, because the distributions are equal for all $ij, rs$.

Next, for $u \in [m]$, $v \in [n]$, let $n_{uv}(R, C)$ be the number of times that $(u, v)$ is sampled in $R, C$. Notice that $\mathbb{E}_{R,C}[n_{uv}(R,C)] = |\Omega| \pi_R(u) \pi_C(v)$. So, by Lemma A.9,

$$
\mathop{\mathbb{E}}_{R,C}\left[ KL\left( (\widetilde{Q}^{(ij)}|R,C), (\widetilde{Q}^{(rs)}|R,C) \right) \right] = \mathop{\mathbb{E}}_{R,C}\left[ \sum_{u \in [m], v \in [n]} \frac{n_{uv}(R,C)}{\sigma_Q^2} (Q_{uv}^{(ij)} - Q_{uv}^{(rs)})^2 \right]
$$
$$
= \mathop{\mathbb{E}}_{R,C}\left[ \frac{\gamma^2}{\sigma_Q^2} (n_{ij}(R,C) + n_{rs}(R,C)) \right]
$$
$$
= \frac{\gamma^2 |\Omega|}{\sigma_Q^2} (\pi_R(i)\pi_C(j) + \pi_R(r)\pi_C(s))
$$

Hence, the average KL divergence for all pairs is:

$$
\frac{1}{d^4} \sum_{(i,j) \in [d]^2} \sum_{(r,s) \in [d]^2} KL(\widetilde{Q}^{(ij)}, \widetilde{Q}^{(rs)}) = \frac{\gamma^2 |\Omega|}{\sigma_Q^2 d^4} \sum_{(i,j) \in [d]^2} \sum_{(r,s) \in [d]^2} (\pi_R(i)\pi_C(j) + \pi_R(r)\pi_C(s))
$$
$$
\leq \frac{\gamma^2 |\Omega|}{\sigma_Q^2 d^4} \sum_{(i,j) \in [d]^2} (1 + d^2 \pi_R(i)\pi_C(j))
$$
$$
\leq \frac{\gamma^2 |\Omega|}{\sigma_Q^2 d^4} \cdot 2d^2
$$
$$
= \frac{2\gamma^2 |\Omega|}{\sigma_Q^2 d^2}
$$

Let $\gamma^2 = \frac{1}{10} \frac{\sigma_Q^2 d^2}{|\Omega|}$. By Theorem A.8, we conclude that for $d \geq 2$, the minimax rate of estimation is at least $\frac{1}{10} \gamma^2 = \frac{1}{100} \frac{\sigma_Q^2 d^2}{|\Omega|}$ $\quad \square$

## A.5. Proof of Proposition 2.4

We use the classical characterization of $G$-optimal designs due to Kiefer and Wolfowitz.

**Theorem A.10** ((Kiefer and Wolfowitz, 1960)). *Let $\pi$ be a distribution on a finite space $\mathcal{A} \subset \mathbb{R}^d$. The following are equivalent:*

- *$\pi$ is $G$-optimal.*

- *$g(\pi) = d$.*

- *For $V(\pi) := \sum_{\boldsymbol{a} \in \mathcal{A}} \pi(\boldsymbol{a}) \boldsymbol{a} \boldsymbol{a}^T$, $\pi$ maximizes $\log \det V(\pi)$.*

We now prove the tensorization of $G$-optimal designs.

**Proposition A.11** (Restatement of Proposition 2.4). *Let $\rho$ be a $G$-optimal design for $\{\hat{U}_P^T \boldsymbol{e}_i : i \in [m]\}$ and $\zeta$ be a $G$-optimal design for $\{\hat{V}_P^T \boldsymbol{e}_j : j \in [n]\}$. Let $\pi(i,j) = \rho(i)\zeta(j)$ be a distribution on $[m] \times [n]$. Then $\pi$ is a $G$-optimal design on $\{\hat{V}_P^T \boldsymbol{e}_j \otimes U_P^T \boldsymbol{e}_i : i \in [m]\}$.*

*Proof.* Let $i \in [m], j \in [n]$. Then by the Kiefer-Wolfowitz theorem,

$$
\begin{aligned}
g(\pi) &= \max_{i,j} \left[ (\hat{V}_P^T e_j \otimes \hat{U}_P^T e_i)^T \left( \sum_{i,j} \pi(i,j)(\hat{V}_P^T e_j \otimes \hat{U}_P^T e_i)(\hat{V}_P^T e_j \otimes \hat{U}_P^T e_i)^T \right)^{-1} (\hat{V}_P^T e_j \otimes \hat{U}_P^T e_i) \right] \\
&= \max_{i,j} \left[ (\hat{V}_P^T e_j \otimes \hat{U}_P^T e_i)^T \left( \left( \sum_j \zeta(j) \hat{V}_P^T e_j e_j^T \hat{V}_P \right) \otimes \left( \sum_i \rho(i) \hat{U}_P^T e_i e_i^T \hat{U}_P \right) \right)^{-1} (\hat{V}_P^T e_j \otimes \hat{U}_P^T e_i) \right] \\
&= \max_{i,j} \left[ (\hat{V}_P^T e_j \otimes \hat{U}_P^T e_i)^T \left[ \left( \sum_j \zeta(j) \hat{V}_P^T e_j e_j^T \hat{V}_P \right)^{-1} \otimes \left( \sum_i \rho(i) \hat{U}_P^T e_i e_i^T \hat{U}_P \right)^{-1} \right] (\hat{V}_P^T e_j \otimes \hat{U}_P^T e_i)^T \right] \\
&= \max_{i,j} \left[ (\hat{V}_P^T e_j)^T \left( \sum_j \zeta(j) \hat{V}_P^T e_j e_j^T \hat{V}_P \right)^{-1} (\hat{V}_P^T e_j)(\hat{U}_P^T e_i)^T \left( \sum_i \rho(i) \hat{U}_P^T e_i e_i^T \hat{U}_P \right)^{-1} (\hat{U}_P^T e_i) \right] \\
&= g(\rho)g(\zeta) \\
&= d^2
\end{aligned}
$$

Where the last step follows from $G$-optimality of $\rho$ and $\zeta$. By Theorem A.10, $\pi$ is $G$-optimal. $\qquad\square$

## A.6. Proof of Theorem 2.6

We first prove a useful error decomposition.

**Proposition A.12** (Decomposition). *Let $\hat{U}_P \in \mathcal{O}^{m \times d}, \hat{V}_P \mathcal{O}^{n \times d}$ be the estimates of the left/right singular vectors of $P$. Then there exist matrices $W_U, W_V \in O(d, \mathbb{R})$ such that if $T_1, T_2$ are the distribution shift matrices as in Definition 1.2, and if $M = (W_U^T T_1) R(T_2^T W_V)$, then:*

$$
Q = \hat{U}_P (W_U^T T_1) R(T_2^T W_V) \hat{V}_P^T + E
$$

*Where the $E$-error depends on the estimator error of $\hat{P}$.*

$$
E := (\hat{U}_P - U_P W_U) M \hat{V}_P^T + \hat{U}_P M (\hat{V}_P - V_P W_V)^T + (\hat{U}_P - U_P W_U) M (\hat{V}_P - V_P W_V)^T
$$

*Proof.* Let $T_1, T_2 \in \mathbb{R}^{d \times d}$ be the distributional shift matrices from Definition 1.2 such that $U_Q = U_P T_1, V_Q = V_P T_2$.

Let $W_U$ be the solution to the Procrustes problem:

$$
W_U := \arg \inf_{W \in \mathcal{O}^{d \times d}} \| U_P W - \hat{U}_P \|_{2 \to \infty}
$$

And similarly,

$$
W_V := \arg \inf_{W \in \mathcal{O}^{d \times d}} \| V_P W - \hat{V}_P \|_{2 \to \infty}
$$

Next, let $Z = T_1 R T_2^T$ and $M = W_U^T Z W_V$. Further, let $\Delta_U = \hat{U}_P - \hat{U}_P W_U$ and $\Delta_V = \hat{V}_P - \hat{V}_P W_V$. Then, we can write $Q$ as:

$$
\begin{aligned}
Q &= U_P T_1 R(V_P T_2)^T \\
&= U_P Z V_P^T \\
&= U_P W_U W_U^T Z W_V W_V^T V_P^T \\
&= (\hat{U}_P + \Delta_U) W_U^T Z W_V (\hat{V}_P + \Delta_V)^T \\
&= \hat{U}_P M \hat{V}_P^T + E
\end{aligned}
$$

Where $E$ contains the cross-terms:

$$
E = \Delta_U M \hat{V}_P^T + \hat{U}_P M \Delta_V^T + \Delta_U M \Delta_V^T
$$

So we are done. $\qquad\square$

We require a strong form of matrix concentration due to (Taupin et al., 2023).

**Lemma A.13** (Design Matrix Concentration). *Let $\hat{\pi}$ be an $\epsilon$-approximate G-optimal design on a finite set $\mathcal{A} \subset \mathbb{R}^d$. Let $\rho, \delta > 0$ and $t \geq 2(1+\epsilon)(\frac{1}{\rho^2} + \frac{1}{3\rho}) d \log(\frac{2d}{\delta})$. Suppose $\Omega = \{a_1, ..., a_t\}$ is the multiset of $t$ samples drawn i.i.d. from $\hat{\pi}$, and let $W_t = \frac{1}{t} \sum_{i=1}^{t} a_i a_i^T$. Then:*

$$\mathbb{P}\left[(1-\rho)\sum_{a \in A} \hat{\pi}(a) a a^T \preceq W_t \preceq (1+\rho)\sum_{a \in A} \hat{\pi}(a) a a^T\right] \geq 1 - \delta$$

*In particular, since $\hat{\pi}$ is $\epsilon$-approximately G-optimal,*

$$\mathbb{P}\left[\frac{d}{(1+\rho)} \leq \max_{a \in A} \|a\|_{W_t^{-1}}^2 \leq \frac{(1+\epsilon)d}{(1-\rho)}\right] \geq 1 - \delta$$

We also require the following standard bound on the maximum of Gaussians.

**Lemma A.14** ((Vershynin, 2018) 2.5.10). *Let $X_1, ..., X_n \overset{i.i.d}{\sim} N(0, \sigma^2)$. Then for all $u > 0$,*

$$\mathbb{P}[\max_i X_i^2 \geq 4\sigma^2 \log(n) + 2u^2] \leq \exp(-\frac{u^2}{2\sigma^2}).$$

*Proof of Theorem 2.6.* We first introduce some notation. Let $S_r, S_c$ be the multisets of rows/columns sampled and $\Omega = S_r \times S_c$. Let $\psi_j = \hat{V}_P^T e_j$ and $\varphi_i = \hat{U}_P^T e_i$. Then, let $\hat{\phi}_{ij} = \hat{V}_P^T e_j \otimes \hat{U}_P^T e_i = \psi_j \otimes \varphi_k$, and $W = \sum_{ij \in \Omega} \hat{\phi}_{ij} \hat{\phi}_{ij}^T$. Notice that:

$$W = \left(\sum_{j \in S_c} \psi_j \psi_j^T\right) \otimes \left(\sum_{i \in S_r} \varphi_i \varphi_i^T\right)$$

Therefore, let $W_1 = \sum_{j \in S_c} \psi_i \psi_j^T$ and $W_2 = \sum_{i \in S_r} \varphi_i \varphi_i^T$ for shorthand. Then $W^{-1}$ exists iff $W_1^{-1}, W_2^{-1}$ exist. By Lemma A.13, both $W_1^{-1}, W_2^{-1}$ exist with probability at least $1 - (m+n)^{-2}$, since $S_r, S_c$ are both large enough by assumption.

Therefore, conditioning on the inverses existing, if we solve the least-squares system, we obtain $\hat{M} \in \mathbb{R}^{d \times d}$ such that:

$$\text{vec}(\hat{M}) = \left(\sum_{ij \in \Omega} \hat{\phi}_{ij} \hat{\phi}_{ij}^T\right)^{-1} \sum_{ij \in \Omega} \hat{\phi}_{ij} \widetilde{Q}_{ij}$$

Recall from Proposition A.12 that $Q = \hat{U}_P M \hat{V}_P^T + E$, where $E_{ij} = \epsilon_{ij}$ is the misspecification error. Therefore, we can bound the error of $\hat{Q} = \hat{U}_P \hat{M} \hat{V}_P^T$ as:

$$\begin{aligned}
\hat{Q}_{ij} - Q_{ij} &= e_i^T \hat{U}_P (\hat{M} - M) \hat{V}_P^T e_j - \epsilon_{ij} \\
&= \hat{\phi}_{ij}^T \text{vec}(\hat{M} - M) + \epsilon_{ij} \\
&=: E_{1;ij} + E_{2;ij} \\
E_{1;ij} &:= \hat{\phi}_{ij}^T \text{vec}(\hat{M} - M) \\
E_{2;ij} &:= \epsilon_{ij}
\end{aligned}$$

Let $G_{ij} \overset{i.i.d}{\sim} N(0,\sigma_Q^2)$ be the additive noise for $\widetilde{Q}_{ij}$. Then, $\widetilde{Q}_{ij} = \hat{\phi}_{ij}^T \text{vec}(M) + \epsilon_{ij} + G_{ij}$. Hence we can write $E_1$ as:

$$
\begin{aligned}
E_{1;k\ell} &= \left( \hat{\phi}_{k\ell}^T (\sum_{ij\in\Omega} \hat{\phi}_{ij}\hat{\phi}_{ij}^T)^{-1} \sum_{ij\in\Omega} \hat{\phi}_{ij}\widetilde{Q}_{ij} \right) - \hat{\phi}_{k\ell}^T \text{vec}(M) \\
&= \hat{\phi}_{k\ell}^T \left( (\sum_{ij\in\Omega} \hat{\phi}_{ij}\hat{\phi}_{ij}^T)^{-1} \sum_{ij\in\Omega} \hat{\phi}_{ij}(\hat{\phi}_{ij}^T \text{vec}(M) + \epsilon_{ij} + G_{ij}) \right) - \hat{\phi}_{k\ell}^T \text{vec}(M) \\
&= \hat{\phi}_{k\ell}^T \left( (\sum_{ij\in\Omega} \hat{\phi}_{ij}\hat{\phi}_{ij}^T)^{-1} \sum_{ij\in\Omega} \hat{\phi}_{ij}(\epsilon_{ij} + G_{ij}) \right) \\
&= \hat{\phi}_{k\ell}^T \left( (\sum_{ij\in\Omega} \hat{\phi}_{ij}\hat{\phi}_{ij}^T)^{-1} \sum_{ij\in\Omega} \hat{\phi}_{ij}\epsilon_{ij} \right) + \hat{\phi}_{k\ell}^T \left( (\sum_{ij\in\Omega} \hat{\phi}_{ij}\hat{\phi}_{ij}^T)^{-1} \sum_{ij\in\Omega} \hat{\phi}_{ij}G_{ij} \right) \\
&=: E_{3;k\ell} + E_{4;k\ell}
\end{aligned}
$$

We analyze $E_4$ first. Let $\boldsymbol{x} = W^{-1} \sum_{ij\in\Omega} \hat{\phi}_{ij}G_{ij}$. For any $k,\ell$, we wish to bound $\hat{\phi}_{k\ell}^T \boldsymbol{x}$. Notice that $\boldsymbol{x}$ is a multivariate Gaussian with mean $\boldsymbol{0}$. Its covariance is therefore:

$$
\mathbb{E}[\boldsymbol{x}\boldsymbol{x}^T] = \sum_{ij\in\Omega}\sum_{i'j'\in\Omega} W^{-1}\hat{\phi}_{ij}\hat{\phi}_{i'j'}^T W^{-1} \mathbb{E}[G_{ij}G_{i'j'}] = \sigma_Q^2 W^{-1}\left(\sum_{ij\in\Omega}\hat{\phi}_{ij}\phi_{ij}^T\right)W^{-1} = \sigma_Q^2 W^{-1}
$$

Hence $\hat{\phi}_{k\ell}^T \boldsymbol{x}$ is a scalar Gaussian with mean zero and variance $\hat{\phi}_{k\ell}^T \sigma_Q^2 W^{-1}\hat{\phi}_{k\ell}$. We next bound this quadratic form. Notice that we can tensorize the quadratic form as:

$$
\begin{aligned}
\phi_{k\ell}^T W^{-1}\phi_{k\ell} &= (\boldsymbol{\psi}_\ell \otimes \boldsymbol{\varphi}_k)^T (W_1 \otimes W_2)^{-1}(\boldsymbol{\psi}_\ell \otimes \boldsymbol{\varphi}_k) \\
&= (\boldsymbol{\psi}_\ell W_1^{-1}\boldsymbol{\psi}_\ell)(\boldsymbol{\varphi}_k W_2^{-1}\boldsymbol{\varphi}_k)
\end{aligned}
$$

We apply Lemma A.13 to each term in the product. With probability $1 - 2(m+n)^{-2}$, for $S_r, S_c$ both of size at least $20d\log(\frac{2d}{m+n})$,

$$
\|\psi_\ell\|_{W_1^{-1}}^2 \|\varphi_k\|_{W_2^{-1}}^2 \leq \frac{(2+2\epsilon)d^2}{|S_r||S_c|}
$$

Conditioning on this event, the variance of $\hat{\phi}_{k\ell}^T \boldsymbol{x}$ is at most $\frac{(1+\epsilon)d^2\sigma_Q^2}{|\Omega|(1-\rho)}$, for $|\Omega| = |S_r||S_c|$. Therefore, by Lemma A.14,

$$
\mathbb{P}\left[ \max_{k\in[m],\ell\in[n]} \left|\hat{\phi}_{k\ell}^T \boldsymbol{x}\right|^2 \leq 20\log(mn)\frac{(2+2\epsilon)\sigma_Q^2 d^2}{|\Omega|} \right] \leq \delta + (mn)^{-2}
$$

Finally, we analyze the error term $E_{3;k\ell}$. By the Cauchy-Schwarz inequality,

$$
|E_{3;k\ell}| \leq \left(\sum_{ij\in\Omega} a_{ij}^2\right)^{1/2}\left(\sum_{ij\in\Omega} \epsilon_{ij}^2\right)^{1/2}
$$

First,

$$
\begin{aligned}
\sum_{ij\in\Omega} a_{ij}^2 &= \sum_{ij\in\Omega} \hat{\phi}_{ij}^T W^{-1} \hat{\phi}_{k\ell} \hat{\phi}_{k\ell}^T W^{-1} \hat{\phi}_{ij} \\
&= \sum_{ij\in\Omega} \mathrm{tr}\left( \hat{\phi}_{ij} \hat{\phi}_{ij}^T W^{-1} \hat{\phi}_{k\ell} \hat{\phi}_{k\ell}^T W^{-1} \right) \\
&= \mathrm{tr}\left( \sum_{ij\in\Omega} \hat{\phi}_{ij} \hat{\phi}_{ij}^T W^{-1} \hat{\phi}_{k\ell} \hat{\phi}_{k\ell}^T W^{-1} \right) \\
&= \mathrm{tr}\left( \hat{\phi}_{k\ell} \hat{\phi}_{k\ell}^T W^{-1} \right) \\
&= \left| \hat{\phi}_{k\ell}^T W^{-1} \hat{\phi}_{k\ell} \right| \\
&\leq \frac{(2+2\epsilon)d^2}{|\Omega|}
\end{aligned}
$$

For the other term,

$$
\Big(\sum_{ij\in\Omega} \epsilon_{ij}^2\Big)^{1/2} \leq |\Omega|^{1/2} \max_{ij\in\Omega} |\epsilon_{ij}|
$$

It follows that $\max_{k,\ell}|E_{3;k\ell}| \leq \sqrt{2+2\epsilon} \cdot d \max_{i,j\in\Omega}|\epsilon_{ij}|$. The conclusion follows. □

### A.7. Proof of Theorem 2.9

We require the following concentration result to control the sizes of masks.

**Lemma A.15** (Bernoulli Concentration). *Let* $X_1,\ldots,X_n \overset{i.i.d}{\sim} Bernoulli(p)$ *for* $p\in(0,1)$. *Then if* $p\geq 10\log n$,

$$
\mathbb{P}\Big[\Big|\sum_i (X_i-p)\Big| \geq \frac{np}{2}\Big] \leq n^{-\omega(1)}
$$

*Proof.* By the scalar Bernstein inequality (Lemma A.16), we have for $B=1$ and $\zeta=np$ that:

$$
\mathbb{P}\Big[\Big|\sum_i (X_i-p)\Big| \geq \tau\Big] \leq 2\exp\Big(-\frac{\tau^2/2}{\zeta+(B\tau/3)}\Big)
$$

Let $\tau=np/2$. Then

$$
\begin{aligned}
\mathbb{P}\Big[\Big|\sum_i (X_i-p)\Big| \geq \tau\Big] &\leq 2\exp\Big(\frac{-10}{8}\log n\Big) \\
&\leq 2n^{-(\log n)^{1/4}}
\end{aligned}
$$

□

We are ready to prove the estimation error for passive sampling.

*Proof of Theorem 2.9.* Following the notation of the proof of Theorem 2.6, we want to bound $E_{3;k\ell}$ and $E_{4;k\ell}$. However, rather than using $G$-optimality to bound quadratic forms of the type $\hat{\phi}_{k\ell} W^{-1} \hat{\phi}_{ij}$, we will apply spectral concentration via Proposition A.17.

To this end, we condition on the events that $\hat{V}_P^T \Pi_R \hat{V}_P \succeq \frac{p_{\text{Row}}}{2}$ and $\hat{U}_P^T \Pi_C \hat{U}_P \succeq \frac{p_{\text{Col}}}{2}$. By Proposition A.4 and Proposition A.17, the two events occur simultaneously with probability $\geq 1 - 2(m \wedge n)^{-10}$. Then $W^{-1}$ exists and $W^{-1} \preceq \frac{4}{p_{\text{Row}}p_{\text{Col}}} I$. Therefore, for all $i, j, k, \ell$, by incoherence,

$$\left| \hat{\phi}_{k\ell}^T W^{-1} \hat{\phi}_{ij} \right| \leq \frac{4}{p_{\text{Row}}p_{\text{Col}}} \|\hat{\phi}_{k\ell}\| \|\hat{\phi}_{ij}\|$$

$$= \frac{4}{p_{\text{Row}}p_{\text{Col}}} \|\boldsymbol{\varphi}_k\| \|\boldsymbol{\varphi}_i\| \|\boldsymbol{\psi}_\ell\| \|\boldsymbol{\psi}_j\|$$

$$\leq \frac{4}{p_{\text{Row}}p_{\text{Col}}} \left( \sqrt{\frac{\mu_U^2 \mu_V^2 d^4}{m^2 n^2}} \right)$$

$$= \frac{4}{p_{\text{Row}}p_{\text{Col}}} \frac{\mu d^2}{mn}$$

Hence, by Lemma A.14,

$$\mathbb{P} \left[ \max_{k \in [m], \ell \in [n]} |E_{4;k\ell}|^2 \leq 20 \log(mn) \sigma_Q^2 \frac{4}{p_{\text{Row}}p_{\text{Col}}} \frac{\mu d^2}{mn} \right] \leq 2(m \wedge n)^{-10} + (mn)^{-2}.$$

Next, we analyze $E_3$. Let $a_{ij} = \hat{\phi}_{k\ell}^T W^{-1} \hat{\phi}_{ij}$. Let $p = q = 2$. By the Cauchy-Schwarz inequality,

$$|E_{3;k\ell}| \leq \left( \sum_{ij \in \Omega} a_{ij}^p \right)^{1/p} \left( \sum_{ij \in \Omega} \epsilon_{ij}^q \right)^{1/q}$$

First, we have:

$$\sum_{ij \in \Omega} a_{ij}^2 = \sum_{ij \in \Omega} \hat{\phi}_{ij}^T W^{-1} \hat{\phi}_{k\ell} \hat{\phi}_{k\ell}^T W^{-1} \hat{\phi}_{ij}$$

$$= \sum_{ij \in \Omega} \text{tr}\left( \hat{\phi}_{ij} \hat{\phi}_{ij}^T W^{-1} \hat{\phi}_{k\ell} \hat{\phi}_{k\ell}^T W^{-1} \right)$$

$$= \text{tr}\left( \sum_{ij \in \Omega} \hat{\phi}_{ij} \hat{\phi}_{ij}^T W^{-1} \hat{\phi}_{k\ell} \hat{\phi}_{k\ell}^T W^{-1} \right)$$

$$= \text{tr}\left( \hat{\phi}_{k\ell} \hat{\phi}_{k\ell}^T W^{-1} \right)$$

$$= \left| \hat{\phi}_{k\ell}^T W^{-1} \hat{\phi}_{k\ell} \right|$$

$$\leq \frac{4\mu d^2}{p_{\text{Row}}p_{\text{Col}}mn}$$

On the other hand,

$$\left( \sum_{ij \in \Omega} \epsilon_{ij}^q \right)^{1/2} \leq |\Omega|^{1/2} \max_{ij \in \Omega} |\epsilon_{ij}|$$

Notice $\mathbb{E}[|\Omega|] = mn p_{\text{Row}} p_{\text{Col}}$. By Lemma A.15, with probability $\geq 1 - 4(m \wedge n)^{-\omega(1)}$,

$$|\Omega| \leq \frac{9}{4} p_{\text{Row}} p_{\text{Col}} mn$$

Therefore, with probability $\geq 1 - 4(m \wedge n)^{-2}$,

$$\frac{\sqrt{|\Omega|}}{p_{\text{Row}}p_{\text{Col}}mn} \leq \frac{3}{2} \frac{1}{\sqrt{p_{\text{Row}}p_{\text{Col}}mn}}$$

The conclusion follows. □

### A.8. Proof of Proposition 2.11

We require the following version of the Matrix Bernstein Inequality (Chen et al., 2021).

**Lemma A.16** (Matrix Bernstein Inequality). *Suppose that $\{Y_i : i = 1, \ldots, n\}$ are independent mean-zero random matrices of size $d_1 \times d_2$, such that $\|Y_i\|_2 \leq B$ almost surely for all $i$, and $\zeta \geq \max\{\|\mathbb{E}[\sum_i Y_i Y_i^T]\|_2, \|\mathbb{E}[\sum_i Y_i^T Y_i]\|_2\}$. Then,*

$$\mathbb{P}\left[\left\|\sum_{i=1}^n Y_i\right\|_2 \geq \tau\right] \leq (d_1 + d_2)\exp\left(-\frac{\tau^2/2}{\zeta + B\tau/3}\right)$$

We now prove nondegeneracy of masks with high probability.

**Proposition A.17** (Spectral Concentration). *Suppose that $\hat{V}_P$ and $\hat{U}_P$ are $\mu_V, \mu_U$-incoherent respectively. Let $\Pi_C \in \{0,1\}^{n \times n}$ be the random matrix with diagonal entries $\nu_1, \ldots, \nu_n$ and similarly let $\Pi_R \in \{0,1\}^{m \times m}$ have diagonal entries $\eta_1, \ldots, \eta_m$. Then, assuming that $\mu_V \leq \frac{p_{\mathrm{Col}} n}{400 d \log n}$ and $\mu_U \leq \frac{p_{\mathrm{Row}} n}{400 d \log n}$, we have:*

$$\mathbb{P}[\hat{U}_P^T \Pi_R \hat{U}_P \succeq p_{\mathrm{Row}}/2] \geq 1 - m^{-10}$$
$$\mathbb{P}[\hat{V}_P^T \Pi_C \hat{V}_P \succeq p_{\mathrm{Col}}/2] \geq 1 - n^{-10}$$

*Proof.* Suppose that $\hat{V}_P$ has rows $\boldsymbol{y}_1, \ldots, \boldsymbol{y}_n \in \mathbb{R}^d$. Then,

$$\hat{V}_P^T \Pi_C \hat{V}_P = \sum_{i=1}^n \nu_i \boldsymbol{y}_i \boldsymbol{y}_i^T.$$

Let $\boldsymbol{v}_i = \sqrt{n}\boldsymbol{y}_i$. Let $p_{\mathrm{Col}} = \mathbb{E}[\nu_i]$. We use $p = p_{\mathrm{Col}}$ for shorthand. Notice $\mathbb{E}[\hat{V}_P^T \Pi_C \hat{V}_P] = \sum_i p \boldsymbol{y}_i \boldsymbol{y}_i^T = p I_d$, since $\hat{V}_P^T \hat{V}_P = I_d$. Therefore,

$$\left\|\sum_i \nu_i \boldsymbol{v}_i \boldsymbol{v}_i^T - pn I_d\right\|_2 = \left\|\sum_i (\nu_i - p)\boldsymbol{v}_i \boldsymbol{v}_i^T\right\|_2$$

Let $Y_i = (\nu_i - p)\boldsymbol{v}_i \boldsymbol{v}_i^T$. Note that $\mathbb{E}[Y_i] = 0$. Next, let $\mu := \mu_V$. By incoherence, $\|Y_i\|_2 \leq \|\boldsymbol{v}_i\|_2^2 \leq \mu d$ for all $i$. Further,

$$\max\{\|\mathbb{E}[\sum_i Y_i Y_i^T]\|_2, \|\mathbb{E}[\sum_i Y_i^T Y_i]\|_2\} = \|\mathbb{E}[\sum_i Y_i^2]\|_2$$

$$= p(1-p)\|\sum_i \|\boldsymbol{v}_i\|_2^2 \boldsymbol{v}_i \boldsymbol{v}_i^T\|_2$$

$$\leq p(1-p)n\mu d\|\sum_i \boldsymbol{y}_i \boldsymbol{y}_i^T\|_2$$

$$= p(1-p)n\mu d$$

Thus, by Lemma A.16, for $B = \mu d$ and $\zeta = p(1-p)n\mu d$, we have:

$$\mathbb{P}\left[\left\|\sum_{i=1}^n Y_i\right\|_2 \geq \tau\right] \leq 2n\exp\left(-\frac{\tau^2/2}{\zeta + B\tau/3}\right)$$

Setting $\tau = 10\sqrt{p(1-p)n\mu d\log n} \vee 10\mu d\sqrt{\log n}$ implies that:

$$\mathbb{P}\left[\sum_i \nu_i \boldsymbol{v}_i \boldsymbol{v}_i^T \succeq pn - \tau\right] \geq 1 - n^{-10}$$

If $\mu \leq \frac{pn}{400 d \log n}$, then $\tau \leq pn/2 = p_{\mathrm{Col}} \cdot n/2$. We conclude that $\mathbb{P}[\hat{V}_P^T \Pi_C \hat{V}_P \succeq p_{\mathrm{Col}}/2] \geq 1 - n^{-10}$. An identical argument gives $\mathbb{P}[\hat{U}_P^T \Pi_R \hat{U}_P \succeq p_{\mathrm{Row}}/2] \geq 1 - m^{-10}$. $\square$

**Corollary A.18.** *Under the assumptions of Proposition A.17, the design matrix for passive sampling has rank $d^2$ with probability at least $1-2(m\wedge n)^{-10}$.*

*Proof.* Let $\Omega \subset [m]\times[n]$ be the set of indices corresponding to the observed entries of $\widetilde{Q}$. Let $P_\Omega \in \{0,1\}^{|\Omega|\times mn}$ be the coordinate projection. The design matrix is precisely $P_\Omega(\hat{V}_P\otimes\hat{U}_P)$. Then, notice that:

$$\big(P_\Omega(\hat{V}_P\otimes\hat{U}_P)\big)^T\big(P_\Omega(\hat{V}_P\otimes\hat{U}_P)\big) = (\hat{V}_P\otimes\hat{U}_P)^T P_\Omega^T P_\Omega(\hat{V}_P\otimes\hat{U}_P)$$
$$= (\hat{V}_P\otimes\hat{U}_P)^T(\Pi_C\otimes\Pi_R)(\hat{V}_P\otimes\hat{U}_P)$$
$$= \hat{V}_P^T\Pi_C\hat{V}_P\otimes\hat{U}_P^T\Pi_R\hat{U}_P$$

By Proposition A.17, this matrix has rank at least $d^2$ with probability $\geq 1-2(m\wedge n)^{-10}$. $\qquad\square$

### A.9. Proof of Theorem 2.12

We require the the Gilbert-Varshamov code (Guruswami et al., 2019).

**Theorem A.19** (Gilbert-Varshamov)**.** *Let $q\geq 2$ be a prime power. For $0<\epsilon<\frac{q-1}{q}$ there exists an $\epsilon$-balanced code $C\subset\mathbb{F}_q^n$ with rate $\Omega(\epsilon^2 n)$.*

We will use the following version of Fano's inequality.

**Theorem A.20** (Generalized Fano Method, (Yu, 1997))**.** *Let $\mathcal{P}$ be a family of probability measures, $(\mathcal{D},d)$ a pseudo-metric space, and $\theta:\mathcal{P}\to\mathcal{D}$ a map that extracts the parameters of interest. For a distinguished $P\in\mathcal{P}$, let $X\sim P$ be the data and $\hat\theta:=\hat\theta(X)$ be an estimator for $\theta(P)$.*

*Let $r\geq 2$ and $\mathcal{P}_r\subset\mathcal{P}$ be a finite hypothesis class of size $r$. Let $\alpha_r,\beta_r>0$ be such that for all $i\neq j$, and all $P_i,P_j\in\mathcal{P}_r$,*

$$d(\theta(P_i),\theta(P_j))\geq\alpha_r;$$
$$KL(P_i,P_j)\leq\beta_r.$$

*Then*

$$\max_{j\in[r]}\mathbb{E}_{P_j}[d(\hat\theta(X),\theta(P_j))]\geq\frac{\alpha_r}{2}\left(1-\frac{\beta_r+\log 2}{\log r}\right).$$

We can now prove Theorem 2.12.

*Proof of Theorem 2.12.* Let $C\subset\{0,1\}^{d^2}$ be the 0.1-balanced Gilbert-Varshmaov code as in Theorem A.19. Let $U,V\in\mathbb{R}^{n\times d}$ be Stiefel matrices with incoherence parameter $\mu=O(1)$. Let $P=U\Sigma_P V^T$ for a diagonal $\Sigma_P\succ 0$ to be specified later. Let $\delta_Q>0$ be a positive real to be specified later.

We will construct a family of source/target pairs indexed by $C$ similar to (Jalan et al., 2024). For $w\in C$, let $B_w\in\mathbb{R}^{d\times d}$ be defined as:

$$B_{w;ij}:=\begin{cases}\frac{\sqrt{mn}}{2d} & w_{ij}=0\\ \frac{\sqrt{mn}}{d}(\frac{1}{2}+\delta_Q) & w_{ij}=1\end{cases}$$

Then define $(P_w,Q_w)=(P,UB_wV^T)$.

For a fixed $w\in C$, the distribution of the data $(A_P,\widetilde{Q})$ depends on the random noise and masking of both $A_P,\widetilde{Q}$. Let $D_R\in\{0,1\}^{m\times m}$ and $D_C\in\{0,1\}^{n\times n}$ be the diagonal matrices corresponding to the row/column masks for $Q$, and let $G\in\mathbb{R}^{m\times n}$ have iid $N(0,\sigma_Q^2)$ entries. Then $\widetilde{Q}=D_R(Q+G)D_C$.

Now, we will apply Theorem A.20 to lower bound $\mathbb{E}\left[\frac{1}{mn}\|\hat{Q}-Q_w\|_F^2\Big|D_R,D_C\right]$. Fix any $D_R\in\text{supp}(\mathcal{E}_1),D_C\in\text{supp}(\mathcal{E}_2)$.

Let $\widetilde{P}_w,\widetilde{Q}_w$ denote the distribution of the data when the population matrices are $P_w,Q_w$ and we condition on the $Q$-mask matrices $D_R,D_C$.

By Theorem A.19, the hypothesis space indexed by $C$ is such that $\log(|C|) \geq C_1 d^2$ for absolute constant $C_1 > 0$. Next, for distinct $w, w' \in C$,

$$KL((\widetilde{P}_w, \widetilde{Q}_w), (\widetilde{P}_{w'}, \widetilde{Q}_{w'})) = KL(\widetilde{P}_{w'}, \widetilde{P}_w) + KL(\widetilde{Q}_w, \widetilde{Q}_{w'})$$
$$\leq KL(\widetilde{Q}_w, \widetilde{Q}_{w'})$$
$$= KL((D_C \otimes D_R)\text{vec}(Q_w + G), (D_C \otimes D_R)\text{vec}(Q_{w'} + G))$$

Notice that we do not use any properties of $\widetilde{P}_w, \widetilde{P}_{w'}$, and in particular allow for deterministic $\widetilde{P}_w = P_w = P$.

Since $D_C, D_R$ are fixed, this is simply the KL divergence of two multiariate Gaussians with the same covariance but different means. Therefore, by Lemma A.9, we have that:

$$KL((\widetilde{P}_w, \widetilde{Q}_w), (\widetilde{P}_{w'}, \widetilde{Q}_{w'})) \leq \frac{1}{\sigma_Q^2} \text{vec}(Q_w - Q_{w'})^T (D_C \otimes D_R)^T (D_C \otimes D_R)^{-1} (D_C \otimes D_R)\text{vec}(Q_w - Q_{w'})$$
$$= \frac{1}{\sigma_Q^2} \| D_R(Q_w - Q_{w'})D_C \|_F^2$$
$$= \frac{1}{\sigma_Q^2} \| D_R U(B_w - B_{w'})V^T D_C \|_F^2$$
$$\leq \frac{1}{\sigma_Q^2} \| D_R U \|_2^2 \| D_C V \|_2^2 \| B_w - B_{w'} \|_F^2$$
$$\leq \frac{5 p_{\text{Row}} p_{\text{Col}}}{\sigma_Q^2} \left( \delta_Q^2 \frac{mn}{d^2} \right) d^2$$
$$= \frac{5 p_{\text{Row}} p_{\text{Col}} mn \delta_Q^2}{\sigma_Q^2}.$$

In the penultimate step, we used the fact that $D_R \in \text{supp}(\mathcal{E}_1), D_C \in \text{supp}(\mathcal{E}_2)$.

Next, for any distinct $w, w' \in C$, by Theorem A.19 we have that $\mathbb{P}_{i,j \in [d]}[w_{ij} \neq w'_{ij}] \geq 0.1$. Therefore,

$$\| Q_w - Q_{w'} \|_F = \| U(B_w - B_{w'})V^T \|_F$$
$$= \| (B_w - B_{w'}) \|_F$$
$$= \left( \sum_{i,j \in [d]: w_{ij} \neq w'_{ij}} \delta_Q^2 \frac{mn}{d^2} \right)^{1/2}$$
$$\geq \frac{1}{10} \delta_Q \sqrt{mn}$$

In the notation of Theorem A.20, we have:

$$\alpha_r := \frac{1}{10} \delta_Q \sqrt{mn}$$
$$\beta_r = \frac{5 p_{\text{Row}} p_{\text{Col}} mn \delta_Q^2}{\sigma_Q^2}$$

Since $\log(|C|) \geq C_1 d^2$, we set $\delta_Q = \sqrt{\frac{C_1 d^2 \sigma_Q^2}{10 p_{\text{Row}} p_{\text{Col}} mn}}$ so that that $\beta_r = \frac{C_1 d^2}{2}$. Therefore, by Theorem A.20, for absolute constants $C_2, C_3, C_4 > 0$,

$$\min_{D_R \in \text{supp}(\mathcal{E}_1), D_C \in \text{supp}(\mathcal{E}_2)} \mathbb{E}\left[ \frac{1}{mn} \| \hat{Q} - Q_w \|_F^2 \,\middle|\, D_R, D_C \right] \geq \frac{C_2 \alpha_r^2}{mn}$$
$$\geq C_3 \delta_Q^2$$
$$\geq \frac{C_4 d^2 \sigma_Q^2}{p_{\text{Row}} p_{\text{Col}} mn}$$

The conclusion follows. □

# B. Additional Experiments and Details

**Compute environment.**   We run all experiments on a Linux machine with 378GB of CPU/RAM. The total compute time across all results in the paper was less than 4 hours.

**Dataset details.**   For the gene expression experiments, we gather whole-blood sepsis gene expression data sampled by (Parnell et al., 2013), available at `https://www.ncbi.nlm.nih.gov/geo/query/acc.cgi?acc=gse54514`. We take the intersection of rows and columns present on days 1 and 2 of the study, and then filter by the 300 most expressed columns (genes) on day 1, to obtain $P, Q \in \mathbb{R}^{31 \times 300}$. Here $P_{ij}$ is the expression level of gene $j$ for patient $i$ on day 1, and $Q_{ij}$ is the same on day 2.

For the metabolic networks experiments, we access the BiGG genome-scale metabolic models datasets (King et al., 2016) at `http://bigg.ucsd.edu`. We use the same set of shared metabolites for iWFL1372 (the source species $P$) and IJN1463 (the target species $Q$) as (Jalan et al., 2024). The resulting networks are weighted undirected graphs with adjacency matrices $P, Q \in \mathbb{R}^{251 \times 251}$ where $P_{ij}$ counts the number of co-occurrences of metabolites $i, j$ in iWFL1372, and $Q_{ij}$ does the same for IJN1463.

**Details of the baselines.**   For the method of (Bhattacharya and Chatterjee, 2022), we use the estimator from their Section 2.2, but modify step (3) to truncate to the true rank $d$, and in step (6) truncate to the true rank of the propensity matrix whose $(i, j)$ entry is $\eta_i \nu_j$. The propensity rank is always 1 in our case. This is the estimator $\hat{Q}_{\text{BC22}} \in \mathbb{R}^{m \times n}$.

For the method of (Levin et al., 2022b), we use the estimator from their Section 3.3, with weights $w_P, w_Q$ based on estimated sub-gamma parameters of the noise for $\widetilde{P}, \widetilde{Q}$. Then, let $Q' \in \mathbb{R}^{m \times n}$ be:

$$Q'_{ij} := \begin{cases} \frac{w_P}{w_P + w_Q} \widetilde{P}_{ij} + \frac{w_Q}{w_P + w_Q} \widetilde{Q}_{ij} & \widetilde{Q}_{ij} \neq \star \\ \widetilde{P}_{ij} & \text{otherwise} \end{cases}$$

We return the rank-$d$ SVD truncation of $Q'$ as $\hat{Q}_{\text{LLL22}} \in \mathbb{R}^{m \times n}$.

We will discuss additional ablation experiments in Section B.2, and experiments on the real-world data in Section B.3.

## B.1. Comparison to the not-MIWAE Method

In this section, we present additional experiments to compare our methods against the *not-MIWAE* method of (Ipsen et al., 2021). Specifically, we compare our active and passive sampling methods on Max Squared Error, Mean Squared Error (MSE), Mean Absolute Error (MAE), and Root Mean Squared Error (RMSE). For ease of comparison, we report the results of Figure 2 (gene expression transfer) and Figure 3 (metabolic transfer) again in the tables below. The MAE/RMSE numbers, and the results of the *not-MIWAE* method, are new.

For the gene expression transfer problem (Figure 2), our methods out-perform *not-MIWAE* with $p_{\text{Row}} = p_{\text{Col}} = 0.5$. We train *not-MIWAE* until convergence, with the latent dimension equal to the true matrix rank of $Q$, and a batch size of 32. For gene expression data, the errors are reported in in Table 3 below.

| Method | MSE | Max Squared Error | MAE | RMSE |
|---|---|---|---|---|
| Passive (Ours) | 0.004385 | 0.300035 | 0.044493 | 0.055198 |
| Active (Ours) | 0.018225 | 0.372105 | 0.103285 | 0.114654 |
| LLL22 | 0.151792 | 0.626293 | 0.343497 | 0.389449 |
| BC22 | 0.570254 | 1.000000 | 0.678862 | 0.754897 |
| not-MIWAE | 0.207850 | 1.000000 | 0.415913 | 0.455765 |

*Table 3.* Performance comparison of different methods in the setting of Figure 2, with $p_{\text{Row}} = p_{\text{Col}} = 0.5$.

Next, we perform the same experiment for the metabolic transfer problem (Figure 3) in Table 4.

Note that our methods may perform better because not-MIWAE is a non-transfer baseline. This further emphasizes the significance of the transfer setting, which our methods capture, as well as the method of (Levin et al., 2022a).

| Method | MSE | Max Squared Error | MAE | RMSE |
|---|---|---|---|---|
| Passive (Ours) | 0.000217 | 1.292995 | 0.000934 | 0.014638 |
| Active (Ours) | 0.000024 | 0.294249 | 0.000669 | 0.004883 |
| LLL22 | 0.000360 | 0.651176 | 0.006931 | 0.018147 |
| BC22 | 0.003790 | 1.000000 | 0.021086 | 0.055543 |
| not-MIWAE | 0.006666 | 1.000000 | 0.030307 | 0.076831 |

*Table 4.* Performance comparison of different methods in the setting of Figure 3, with $p_{\text{Row}} = p_{\text{Col}} = 0.5$.

## B.2. Ablation Studies

Throughout this section we use the Partitioned Matrix Model with $a = 0.1, b = 0.8$ from Section 3. For each setting, we hold all parameters fixed and vary one parameter pto observe the effect of all algorithms on both Max Sqaured Error and Mean Squared Error. The default settings are:

- Matrices $P, Q \in \mathbb{R}^{m \times n}$ with $m = 300, n = 200$.

- The parameters $a = 0.8, b = 0.1$ in the Partitioned Matrix Model.

- Additive noise for $\widetilde{Q}$ is iid $\mathcal{N}(0, \sigma_Q^2)$ with $\sigma_Q = 0.1$.

- The rank is $d = 5$.

- $p_{\text{Row}} = p_{\text{Col}} = 0.5$, so the probability of seeing any entry of $Q$ is $0.25$.

For all experiments, we test for 10 independent trials at each parameter setting and display the median error of each method, along with the $[10, 90]$ percentile.

Figure 9 shows that all methods do poorly in max error when $P$ is masked. Our methods are best in mean-squared error. This is because the Matrix Partition Model is highly coherent, as can be shown from spectral partitioning arguments (Lee et al., 2014). Therefore, the max-squared error is high, as we would expect from Remark 2.7 and the results of (Chen et al., 2020).

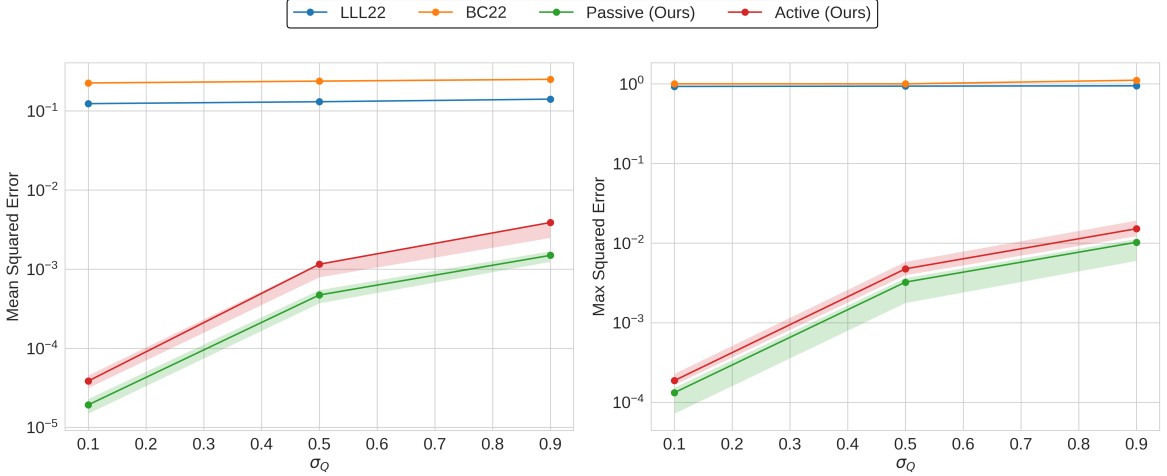

*Figure 5.* We test the effect of growing the target additive noise parameter $\sigma_Q$.

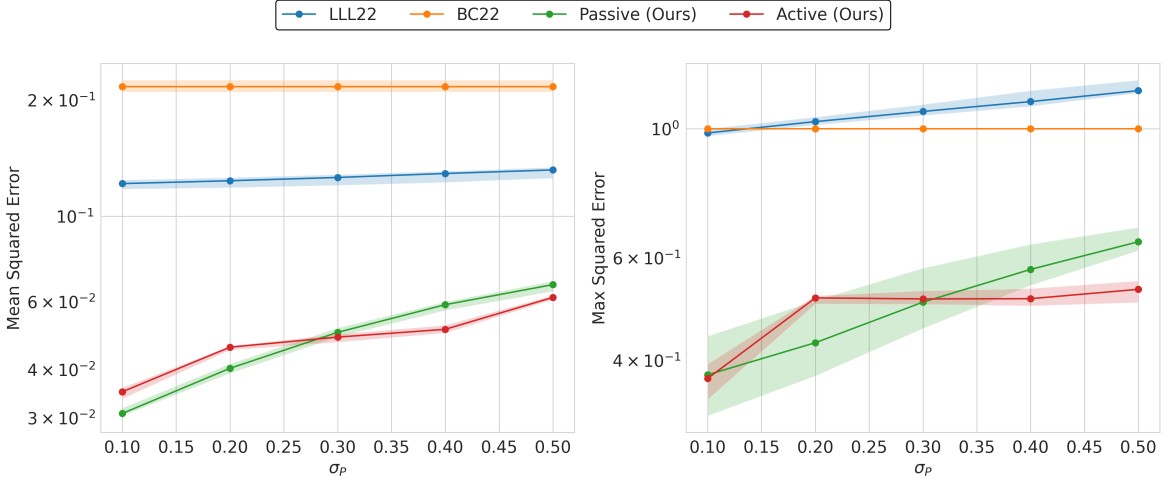

*Figure 6.* We test the effect of growing the target additive noise parameter $\sigma_P$. Each entry of $P$ is observed with i.i.d. additive noise $\mathcal{N}(0, \sigma_P^2)$.

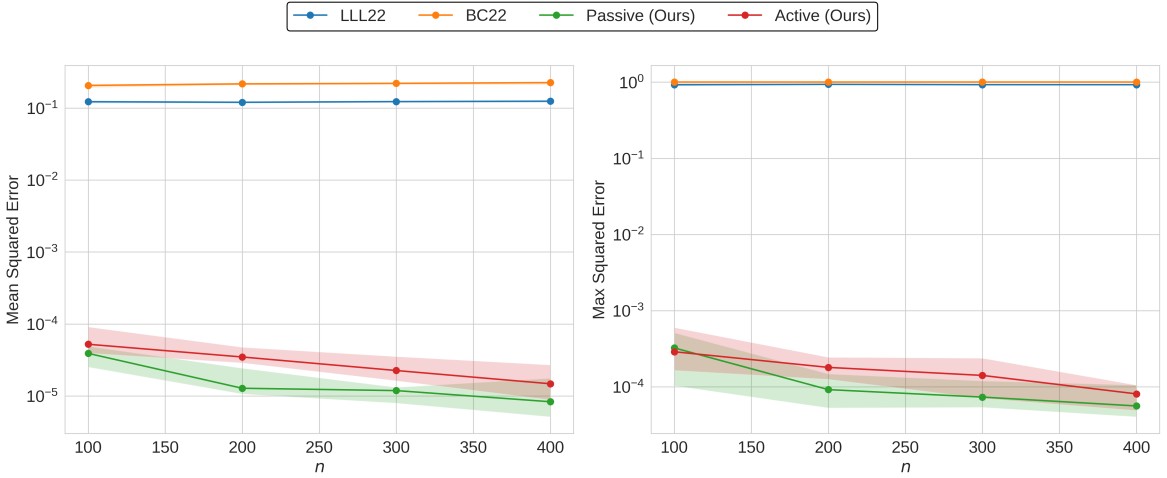

*Figure 7.* We test the effect of growing $n$ for $P, Q \in \mathbb{R}^{300 \times n}$.

### B.3. Additional Real-World Experiments

We first display the weighted adjacency matrices for $P, Q$ for the metabolic networks setting of Section 3 as Figure 10 and Figure 11. It is evident that the edge weights show significant skew. Note that the colorbar for both visualizations is logarithmically scaled.

Next, we report mean-squared error for the same experimental settings discussed in Section 3. Figure 13 shows the results for gene expression. Figure 12 shows the results for metabolic data; notably, despite poor performance in max-squared error, the passive sampling estimator is reasonably good in mean-squared error, although not as good as the active sampling estimator.

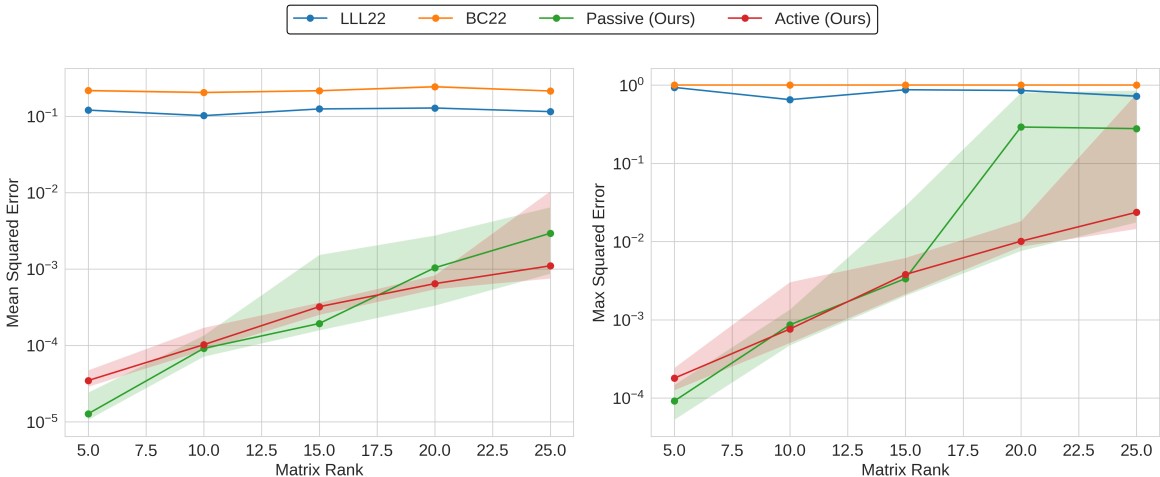

*Figure 8.* We test the effect of rank.

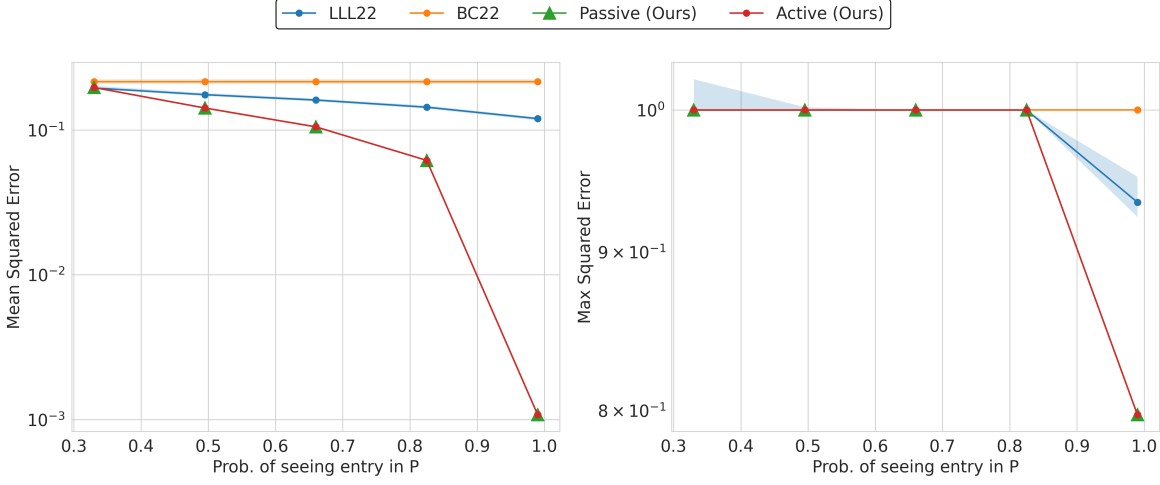

*Figure 9.* We test the effect of masking entries of $P$ in a Missing Completely-at-Random setup with probability $p$. Note that the errors for active and passive sampling are almost identical, so we use different markers (circle and triangle resp.) to distinguish them. We see that our methods do better in mean-squared error (left) while max error is poor for all methods (right).

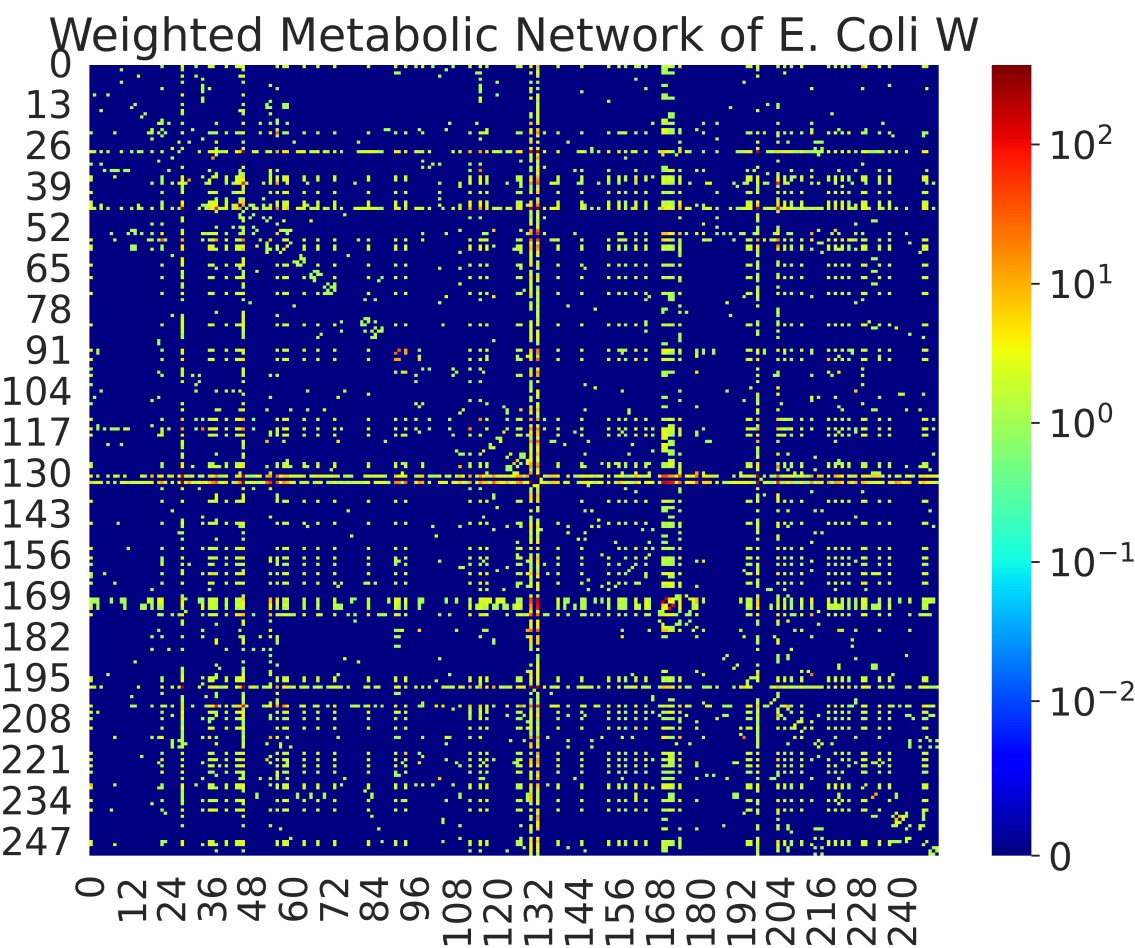

*Figure 10.* The source matrix $P$ in the setting of Figure 3.

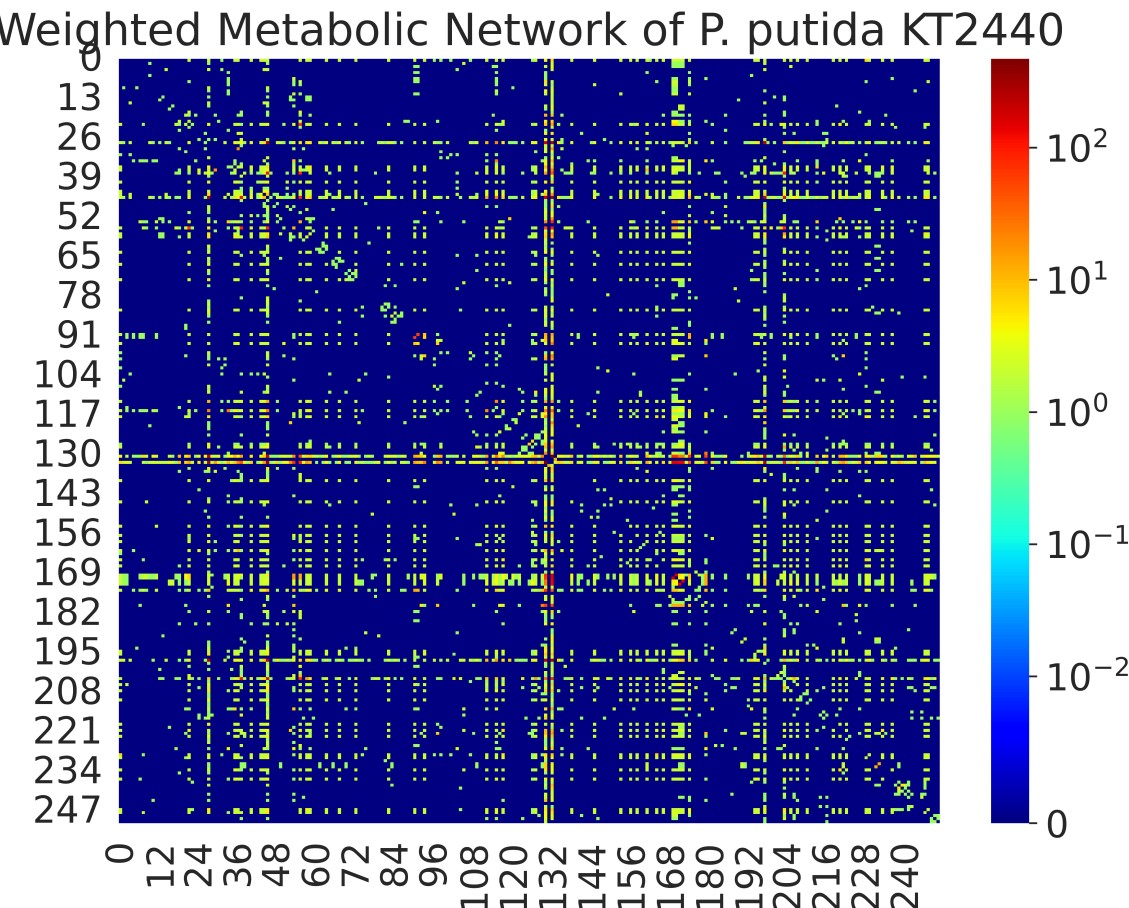

*Figure 11.* The target matrix $Q$ in the setting of Figure 3.

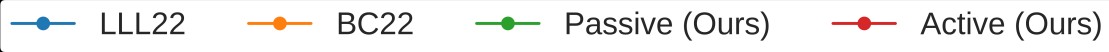

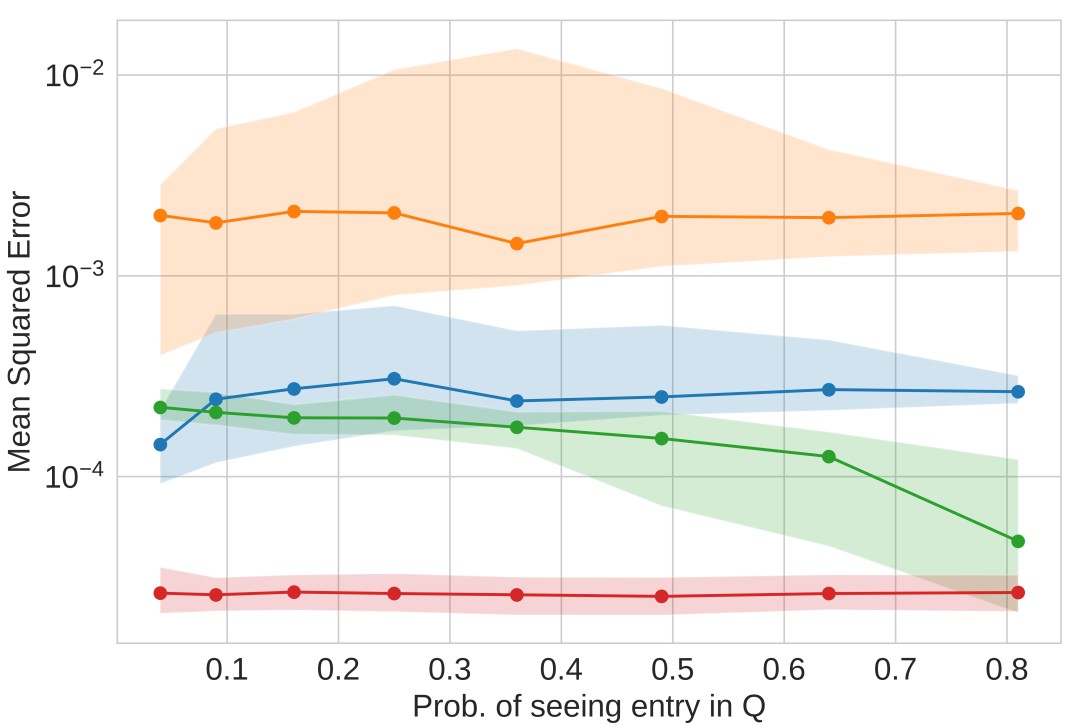

*Figure 12.* The mean-squared error of each $\hat{Q} - Q$ in the setting of Figure 3.

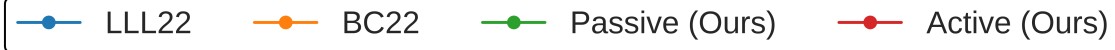

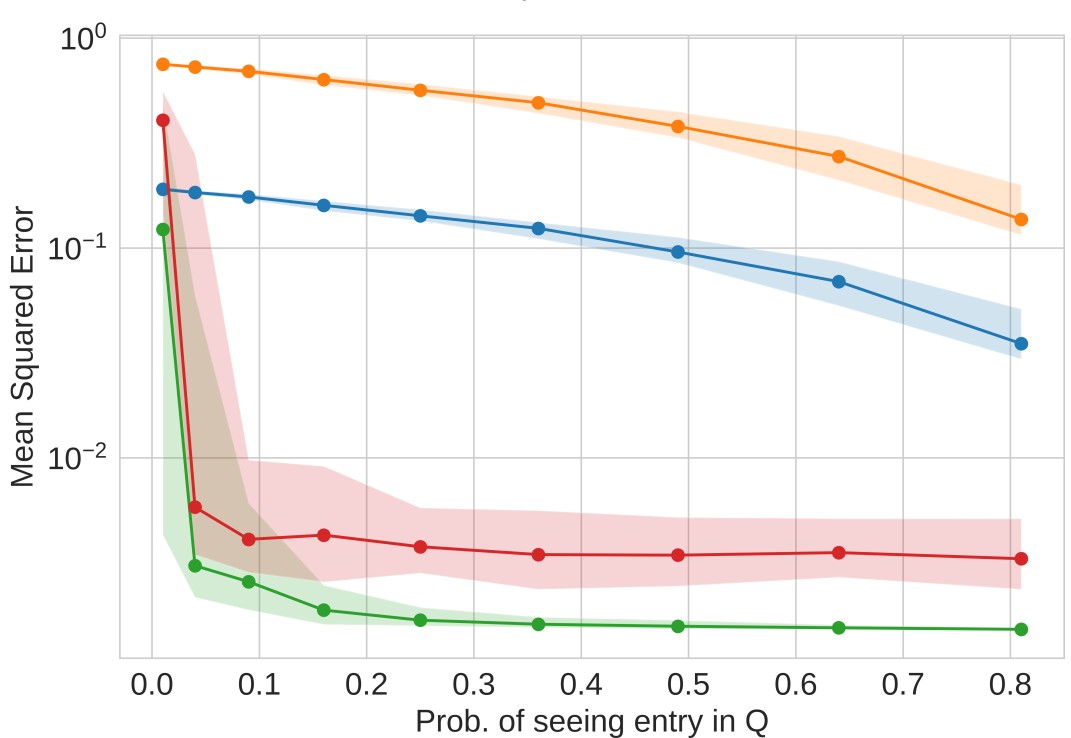

*Figure 13.* The mean-squared error of each $\hat{Q} - Q$ in the setting of Figure 2.

