# OpenReview forum: "Optimal Transfer Learning for Missing Not-at-Random Matrix Completion"
_ICML.cc/2025/Conference — ICML 2025 poster_

### Official Review · Reviewer_nRTL · 2025-03-08

**Overall Recommendation:** 3

**Summary:**

This paper studies the problem of matrix completion with missing not-at-random mechanisms, where the observation pattern is row/columns-wise. Under such missing/observation family, the authors establish the minimax lower bound for entrywise estimation error. With side information, the authors propose a computationally efficient estimation framework which achieves the optimal rate under mild assumptions. The effectiveness of proposed method is demonstrated on both simulated data and real-world genomic and metabolic datasets.

**Claims And Evidence:**

The claims are supported by rigorous theoretical investigations and numerical experiments.

**Essential References Not Discussed:**

No

**Experimental Designs Or Analyses:**

If I understand it correctly, there's no real missing in the real world experiments. Are the missed entries synthetic? It would enhance the significance of this work if the experiments are conducted on datasets where the primary interest is the recovery of entries.

**Methods And Evaluation Criteria:**

The theoretical results are developed mostly in $max$ (entrywise $L_\infty$) norm. It would be interesting to see the discussion in other forms (in an average sense, like Frobenius norm or entrywise $L_1$ norm). In numerical experiments, such metrics are also interesting and informative to see (RMSE, MAE, etc).

**Other Comments Or Suggestions:**

line 1422, "do poorly in max errow"

**Other Strengths And Weaknesses:**

No.

**Questions For Authors:**

No

**Relation To Broader Scientific Literature:**

The presented method works effectiveness for biological findings.

**Theoretical Claims:**

I didn't go through the detailed proofs, but the ideas and discussions presented in the main body are sound to me.

---

> ### Author Rebuttal · Authors · 2025-03-31
>
> We thank the reviewer for their valuable feedback, and for highlighting that our claims are supported by rigorous theoretical investigations and numerical experiments. We will fix typos in the revision, and we address all other comments below.
>
> ## Our methods perform best under new evaluation metrics (MAE and RMSE).
>
> As the Reviewer suggests, we perform new experiments to compare our active and passive sampling methods on Mean Absolute Error (MAE), Root Mean Squared Error (RMSE), as well as the the previously presented metrics of Max Squared Error, and Mean Squared Error (MSE). For ease of comparison, we report the results of Fig. 2 (gene expression transfer) and Fig. 3 (metabolic transfer) again in the tables below. The MAE/RMSE numbers are new.
>
> For the gene expression transfer problem (Figure 2), our methods continue to out-perform baselines with $p_{\textup{Row}} = p_{\textup{Col}} = 0.5$.
> For gene expression data, the errors are:
> | Method | MSE | Max Squared Error | MAE | RMSE |
> |--------|-------|-----------|-----|------|
> | Passive (Ours) | **0.004385** | **0.300035** | **0.044493** | **0.055198** |
> | Active (Ours) | *0.018225* | *0.372105* | *0.103285* | *0.114654* |
> | LLL22 | 0.151792 | 0.626293 | 0.343497 | 0.389449 |
> | BC22 | 0.570254 | 1.000000 | 0.678862 | 0.754897 |
>
> Next, we perform the same experiment for the metabolic transfer problem (Figure 3).
> | Method | MSE | Max Squared Error | MAE | RMSE |
> |--------|-------|-----------|-----|------|
> | Passive (Ours) | *0.000217* | 1.292995 | *0.000934* | *0.014638* |
> | Active (Ours) | **0.000024** | **0.294249** | **0.000669** | **0.004883** |
> | LLL22 | 0.000360 | *0.651176* | 0.006931 | 0.018147 |
> | BC22 | 0.003790 | 1.000000 | 0.021086 | 0.055543 |
>
> We will include the suggested error metrics (MAE and RMSE) in the revision.
>
> ## Our theoretical upper bounds on Max Squared Error also imply upper bounds on Frobenius error, RMSE, and MAE.
>
> The max-squared error bounds (Theorem 2.6 and Theorem 2.9) immediately imply bounds on Frobenius error, because $\frac{1}{mn} \Vert \hat Q - Q \Vert_{F}^2 \leq \Vert \hat Q - Q \Vert_{max}^2$.
>
> Similarly, both the RMSE and the MAE are upper-bounded by the Max Absolute Error, which can be bounded via Theorems 2.6 and 2.9.
>
> We will include this discussion in the revision.
>
> ## Our experiments are conducted on datasets where the primary interest is the recovery of entries.
>
> Our real-world experiments (Section 3.1) on gene expression (Fig. 2) and metabolic (Fig. 3) data are on datasets where the primary interest is recovery of missing entries (Parnell et al. 2013, King et al. 2016, Jalan et al. 2024).
>
> ## We mask matrix entries to compare estimated vs. ground-truth values.
>
> In each experiment, we mask entries of $Q$ for which the ground-truth is known, so that we can compare the estimated $\hat Q_{ij}$ to the ground truth value $Q_{ij}$. We mask according to the active and passive sampling frameworks (lines 302-320, left).
>
> There are many entries of $P, Q$ for which the ground truth is unknown. Without knowing the ground truth, we cannot report estimation error. Therefore we do not perform estimation experiments on missing entries for which ground-truth values are not known.

---

### Official Review · Reviewer_zFpJ · 2025-03-13

**Overall Recommendation:** 3

**Summary:**

This paper studies transfer learning for matrix completion in a Missing Not-at-Random (MNAR) setting, which is motivated by biological problems. The problem is challenging because entire rows and columns of the target matrix are are missing, making direct estimation impossible. This paper introduces a source matrix. This paper provides lower bounds for estimation error in both active and passive sampling settings. The proposed model is an efficient model that is minimax-optimal in the active setting and rate-optimal in the passive setting.

**Claims And Evidence:**

1. The proposed estimator is minimax-optimal for the active setting, while in real-world applications, exact optimization of sample selection is impractical.
2. The transfer learning framework effectively corrects for the missingness structure in MNAR settings. However, the distribution shift model is restrictive.

**Essential References Not Discussed:**

Graph-based and VAE-based matrix completion methods should be discussed.

**Experimental Designs Or Analyses:**

1. The paper evaluates on only two datasets, which is insufficient for demonstrating broad applicability.
2. The paper does not evaluate how well the method generalizes across different datasets or varying missingness patterns.

**Methods And Evaluation Criteria:**

1. The paper defines active and passive sampling settings, but the evaluation assumes clean singular value decomposition (SVD) features can always be extracted, which is unrealistic in noisy biological data.
2. Active sampling relies on an idealized G-optimal design, which assumes one can perfectly select the most informative rows/columns, an unrealistic assumption in biological experiments.

**Other Comments Or Suggestions:**

1. Discuss practical limitations of active sampling and whether it can be applied in real experimental settings.
2. Provide more details on implementation choices, such as hyperparameter tuning and computational complexity.

**Other Strengths And Weaknesses:**

Strengths:
1. The problem of MNAR matrix completion with transfer learning is relevant in biological applications.
2. The theoretical analysis of estimation error bounds contributes to the understanding of MNAR matrix recovery.

Weaknesses:
1. The method is largely a combination of existing techniques.
2. Real-world applicability is questionable, from my perspective, the active sampling strategy is difficult to implement practically.
3. The assumptions about the structured linear feature shifts between source and target matrices is restrictive.

**Questions For Authors:**

1. What happens if the structured transfer assumption (Definition 1.2) does not hold?
2. Can the active sampling method be realistically implemented in biological experiments?
3. How robust is the method to different missingness mechanisms?

**Relation To Broader Scientific Literature:**

This paper studies transfer learning for matrix completion in a Missing Not-at-Random (MNAR) setting, which is motivated by biological problems.

**Theoretical Claims:**

1. Minimax lower bounds for MNAR matrix completion are presented, but the practical significance of these bounds is unclear.
2. The paper proves optimality under restrictive assumptions, while real-world missingness mechanisms are more complex.

---

> ### Author Rebuttal · Authors · 2025-03-31
>
> We thank the reviewer for their valuable feedback, and for highlighting the relevance of our problem (MNAR matrix completion with transfer learning) to biological applications. We address their comments below.
>
> ## Our minimax results guide algorithm design.
>
> The practical significance of our minimax lower bounds for MNAR matrix completion is that *no method* can achieve a better estimation error than ours. So, the error guarantees in Theorems 2.6 and 2.9 are unimprovable in our setting.
>
> ## Exact selection of rows and columns (in active sampling) *is realistic* for the biological settings we study.
>
> The choice of exact rows and columns to query matches experimental design constraints in multiple settings (lines 9-16, right). Our model is designed to capture these settings. These include (i) metabolite balancing experiments (Christensen and Nielsen, 2000), (ii) gene expression microarrays (Hu et al. 2021), (iii) marker selection for single-cell RNA sequencing (Vargo and Gilbert, 2020) and (iv) patient selection for companion diagnostics (Huber et al. 2022). In Section 3.1, we study precisely the first two settings listed in the introduction: metabolic (Fig. 3) and gene expression (Fig. 2) data.
>
> Some settings, such as electronic health records (Zhou et al. 2023), have different experimental design constraints (lines 46-49, right). Our model would not apply to these settings.
>
> In the revision, we will include additional details on experimental protocols for metabolite balancing and gene expression microarray experiments, as suggested.
>
> ## Our MNAR model captures the missingness structure of the biological datasets we study.
>
> Our passive sampling model (lines 97-109, left) matches the row/column missingness structure present in gene expression data (Fig. 1, and lines 16-24, right) and metabolic network data (lines 330-342, left). We focus on these settings in our real-world experiments (Section 3.1).
>
> For other kinds of data, other missigness structures may be present (lines 46-49, right). Our methods can be applied to *any* missingness structure, but we analyze the specific MNAR settings of the paper due to their biological importance.
>
> If the Reviewer has particular missingness structures in mind, we are happy to discuss the applicability of our methods to those settings.
>
> ## The distribution shift model is realistic for the datasets we study.
>
> The matrix transfer model (Definition 1.2) is commonly used in the biological literature, such as in Genome-Wide Association Studies (see McGrath et al. 2024 and references therein). Our real-world experiments (Section 3.1) further validate that it is appropriate for gene expression (Fig. 2) and metabolic (Fig. 3) transfer problems.
>
> No model applies perfectly to real-world data. It would be interesting to study other models of distribution shift, as we have stated (lines 437-439, right).
>
> ## Our methods can be applied without any structured transfer assumptions.
>
> Even if Definition 1.2 does not hold, our method can be applied. However, we only analyze its theoretical properties under Definition 1.2.
>
> ## Our methods *can* use noisy SVD estimates.
>
> We do not assume clean SVD features for either the source matrix $P$ or target matrix $Q$. The source matrix, which follows Assumption 2.5, can be noisily observed in an MCAR or MNAR fashion (lines 167-172, right). The target matrix is observed with additive noise and has missing rows and columns for both active and passive sampling (lines 79-109, left).
>
> ## Our active sampler *does not* assume idealized G-optimal design.
>
> We compute an $\epsilon$-approximate G-optimal design (Definition 2.3) with convex optimization. Specifically, we use the Franke-Wolfe algorithm which runs in polynomial time (Lattimore and Szepesvari 2020). Setting $\epsilon = 0.05$ is sufficient (Theorem 2.6). A perfect $G$-optimal design is not needed.
>
> ## We will add graph-based and VAE-based references in the revision.
>
> We will discuss VAE-based methods and graph-based methods in the revision, including but not limited to the (see discussion with Reviewer XH4Z) of Ipsen et al. 2020, as well as [1-2] for VAE, and Jalan et al. (NeurIPS 2024) and [3-4] for graphs.
>
> [1] Ghalebikesabi, Sahra, et al. "Deep generative missingness pattern-set mixture models." International conference on artificial intelligence and statistics. PMLR, 2021.
>
> [2] Cai, Hongmin, et al. "Realize generative yet complete latent representation for incomplete multi-view learning." IEEE Transactions on Pattern Analysis and Machine Intelligence46.5 (2023): 3637-3652.
>
> [3] Wang, Yao, et al. "Matrix Completion with Graph Information: A Provable Nonconvex Optimization Approach." arXiv preprint arXiv:2502.08536 (2025).
>
> [4] Zhan, Tong, et al. "Collective Matrix Completion via Graph Extraction." IEEE Signal Processing Letters (2024).
>
> ## All algorithms are polynomial-time.
>
> We use well-studied polynomial-time algorithms (SVD, least-squares, and Franke-Wolfe).

---

### Official Review · Reviewer_hEy4 · 2025-03-14

**Overall Recommendation:** 4

**Summary:**

This paper studies a problem in which there are two matrices $P$ and $Q$ of the same dimension, where a noisy version of $P$ is observed and a noisy and partial view of $Q$ is observed. $P$ is known to be low-rank, and the two matrices are related via distribution shift. The objective is to recover the matrix $Q$ using this additional knowledge. The authors study two settings, one in which there is a budget to sample noisy entries of $Q$ (active sampling) and one in which, according to some distribution, noisy entries of $Q$ are revealed. An estimation algorithm for $Q$ is given, and analysis is provided for the estimation error of both settings.

**Claims And Evidence:**

Yes.

**Essential References Not Discussed:**

n/a

**Experimental Designs Or Analyses:**

Yes, the experiments appear to be sound and valid.

**Methods And Evaluation Criteria:**

Yes, and Figure 2 shows that that this method outperforms other methods as well.

**Other Comments Or Suggestions:**

See above.

**Other Strengths And Weaknesses:**

Strengths:
* The paper is comprehensive in its analysis of an interesting problem and studies two different settings, the passive and active sampling settings. As expected, the active sampling setting is easier. Minimax lower bounds and generic error bounds are given for both settings. Further, the solution is computationally efficient.
* This paper generalizes on previous results by allowing for any kind of distribution shift rather than simply a rotational shift.
* Empirically, there are strong results. The results appear to be complete and well-described.
Weaknesses:
* To improve readability, it would be better to first introduce the estimation framework, and then separately discuss active and passive learning settings.

**Questions For Authors:**

n/a

**Relation To Broader Scientific Literature:**

The problem analyzed in this paper is motivated by problems that arise in biology. The authors emphasize its relation to tasks such as various key biology and medical problems, for which this paper may be useful.

**Theoretical Claims:**

The theoretical claims appear to be correct.

---

> ### Author Rebuttal · Authors · 2025-03-31
>
> We thank the reviewer for their valuable feedback, and for highlighting that our paper has strong empirical results, and is comprehensive in its analysis of an interesting problem. We address their comments below.
>
> ## We will reorganize the writing to first introduce the estimation framework, and then separately discuss the active and passive sampling settings.
>
> Our estimation framework (Section 2.2) involves first learning features from the source matrix via SVD, and then estimating the target matrix via least-squares. This estimation framework applies to both the active sampling (lines 85-96, left) and passive sampling (lines 97-109, left) settings. To improve readability, we will reorganize the problem setup in the revision to first introduce the estimation framework, and then discuss both the active and passive sampling settings. We thank the reviewer for this valuable suggestion.

---

### Official Review · Reviewer_XH4Z · 2025-03-19

**Overall Recommendation:** 3

**Summary:**

The authors study matrix completion in the MNAR setting under transfer learning. They establish minimax bounds for entry-wise estimation of target values under both active and passive sampling settings. Additionally, they propose a computationally efficient minimax-optimal estimator—leveraging the tensorization of G-optimal designs for active sampling—and a rate-optimal estimator for passive sampling. Experiments on two simulated datasets and two real-world datasets demonstrate the effectiveness of their method compared to two baseline approaches.

**Claims And Evidence:**

The theoretical results are well established and proved.

**Essential References Not Discussed:**

I think it would the best if the author discuss more about missing data imputation method under MNAR in the literature.

**Experimental Designs Or Analyses:**

In the experimental design, it would be valuable to include results for a non-transfer setting. Specifically, evaluating state-of-the-art imputation methods when trained solely on Q would provide a useful baseline. If the proposed method outperforms these non-transfer baselines, it would further highlight the significance of studying the transfer setting and demonstrate its practical advantages. I encourage the authors to consider this comparison to strengthen their empirical evaluation.

**Methods And Evaluation Criteria:**

The paper reports Mean Squared Error (MSE) for tasks involving gene expression microarrays and metabolic modeling. Max Squared Error (MSE_max) might be commonly used in these domains due to its sensitivity to extreme deviations, which can be biologically significant. It might be helpful for the authors to adopt other evaluation metrics (e.g., Mean Absolute Error (MAE), and Root Mean Squared Error (RMSE) ) for a more comprehensive assessment of model performance.

**Other Comments Or Suggestions:**

None

**Other Strengths And Weaknesses:**

Strengths: The paper is well written and presents a sound theoretical framework.
Weaknesses: The baseline methods in the experimental section are somewhat limited. While I acknowledge that few methods specifically address matrix completion in the transfer setting, it would be beneficial to incorporate baseline methods from the MNAR imputation literature—such as not-MIWAE [1]—to establish a stronger point of comparison. Evaluating these methods using only data from Q would help demonstrate the best possible performance without transfer, thereby further emphasizing the significance of the transfer setting.

[1] Ipsen, N.B., Mattei, P., & Frellsen, J. (2020). not-MIWAE: Deep Generative Modelling with Missing not at Random Data. ArXiv, abs/2006.12871.

**Questions For Authors:**

Refer to Experimental Designs Or Analyses about the evaluation criterion.

**Relation To Broader Scientific Literature:**

I think this paper help to establish the theoretical and practical approach for matrix complete under transfer setting.

**Theoretical Claims:**

Yes, I mainly check the theorem mentioned in the main paper.

---

> ### Author Rebuttal · Authors · 2025-03-31
>
> We thank the reviewer for their valuable feedback, and for highlighting that our paper is well written and presents a sound theoretical framework. We address their comments below.
>
> ## The manuscript contains results for a non-transfer MNAR baseline (BC22).
>
> The method of BC22 (IEEE Transactions on Information Theory, 2022) is an MNAR imputation method (lines 302-305, left). The manuscript includes this method for all experiments.
>
> ## Our methods outperform not-MIWAE on MAE, RMSE, MSE, and Max Error.
>
> As the Reviewer suggests, we perform new experiments to compare our methods against the not-MIWAE method of Ipsen et al. 2020. Below, we present the comparison of our active and passive sampling methods on Max Squared Error, Mean Squared Error (MSE), Mean Absolute Error (MAE), and Root Mean Squared Error (RMSE). For ease of comparison, we report the results of Fig. 2 (gene expression transfer) and Fig. 3 (metabolic transfer) again in the tables below. The MAE/RMSE numbers, and the results of not-MIWAE, are new.
>
> For the gene expression transfer problem (Figure 2), our methods out-perform not-MIWAE with $p_{\textup{Row}} = p_{\textup{Col}} = 0.5$. We train not-MIWAE until convergence, with the latent dimension equal to the true matrix rank of $Q$, and a batch size of 32.
> For gene expression data, the errors are:
> | Method | MSE | Max Squared Error | MAE | RMSE |
> |--------|-------|-----------|-----|------|
> | Passive (Ours) | **0.004385** | **0.300035** | **0.044493** | **0.055198** |
> | Active (Ours) | *0.018225* | *0.372105* | *0.103285* | *0.114654* |
> | LLL22 | 0.151792 | 0.626293 | 0.343497 | 0.389449 |
> | BC22 | 0.570254 | 1.000000 | 0.678862 | 0.754897 |
> | not-MIWAE | 0.207850 | 1.000000 | 0.415913 | 0.455765 |
>
> Next, we perform the same experiment for the metabolic transfer problem (Figure 3).
> | Method | MSE | Max Squared Error | MAE | RMSE |
> |--------|-------|-----------|-----|------|
> | Passive (Ours) | *0.000217* | 1.292995 | *0.000934* | *0.014638* |
> | Active (Ours) | **0.000024** | **0.294249** | **0.000669** | **0.004883** |
> | LLL22 | 0.000360 | *0.651176* | 0.006931 | 0.018147 |
> | BC22 | 0.003790 | 1.000000 | 0.021086 | 0.055543 |
> | not-MIWAE | 0.006666 | 1.000000 | 0.030307 | 0.076841 |
>
> As Reviewer XH4Z suggests, our methods may perform better because not-MIWAE is a non-transfer baseline. This further emphasizes the significance of the transfer setting, which our methods capture, as well as the method of LLL22.
>
> We will include both the suggested baseline (not-MIWAE), and the suggested error metrics (MAE and RMSE) in the revision.
>
> ## Our methods out-perform others under new evaluation metrics (MAE and RMSE).
>
> See the above table. We will include these new metrics in the revision to give a more comprehensive assessment of model performance.
>
> As the reviewer correctly notes, our main focus is on Max Squared Error, which is a commonly used evaluation metric due to its sensitivity to extreme deviations; this is biologically significant.

---

### Decision · Program_Chairs · 2025-05-01

**Decision:**

Accept (poster)

**Comment:**

This paper studies matrix completion in a novel framework that incorporates missingness of entire rows or columns with some side information available in the form of an out-of-domain source matrix. The results include both upper and lower bounds in both passive and active sampling regimes, where the latter refers to a situation where the learner is able to select a certain quota of full rows and columns to observe. Like the "MNAR" literature, the results are "out of distribution" in the sense that the upper bounds manage to control the recovery error in terms of the maximum error on any entry, even in though the sampling distribution is nonuniform.  The reviewers unanimously agree the paper makes a strong contribution and should be accepted. I also find the results very interesting and powerful.  It would be interesting to compare the results to the existing generalization analyses for IMC in the distribution-free  i.i.d. sampling regime, where the performance measure is defined in-distribution (in expectation over the sampling distribution).